# Less Is More, but Where?
# Dynamic Token Compression via LLM-Guided Keyframe Prior

**Yulin Li**[1][*]   **Haokun Gui**[4][*]   **Ziyang Fan**[1]   **Junjie Wang**[1]   **Bin Kang**[3]   **Bin Chen**[1,3][†]
**Zhuotao Tian**[1,2][†]

[1]Harbin Institute of Technology (Shenzhen)   [2]Shenzhen Loop Area Institute
[3]University of Chinese Academy of Sciences   [4]The Hong Kong University of Science and Technology

## Abstract

Recent advances in Video Large Language Models (VLLMs) have achieved remarkable video understanding capabilities, yet face critical efficiency bottlenecks due to quadratic computational growth with lengthy visual token sequences of long videos. While existing keyframe sampling methods can improve temporal modeling efficiency, additional computational cost is introduced before feature encoding, and the binary frame selection paradigm is found suboptimal. Therefore, in this work, we propose **Dy**namic **To**ken compression via LLM-guided **K**eyframe prior (**DyToK**), a training-free paradigm that enables dynamic token compression by harnessing VLLMs' inherent attention mechanisms. Our analysis reveals that VLLM attention layers naturally encoding query-conditioned keyframe priors, by which DyToK dynamically adjusts per-frame token retention ratios, prioritizing semantically rich frames while suppressing redundancies. Extensive experiments demonstrate that DyToK achieves state-of-the-art efficiency-accuracy tradeoffs. DyToK shows plug-and-play compatibility with existing compression methods, such as VisionZip and FastV, attaining $4.3\times$ faster inference while preserving accuracy across multiple VLLMs, such as LLaVA-OneVision and Qwen2.5-VL. Code is available at https://github.com/yu-lin-li/DyToK.

## 1  Introduction

Recent advancements in Video Large Language Models (VLLMs) [1, 2, 3, 4] have demonstrated remarkable capabilities in processing complex video content. However, their practical deployment faces critical challenges when handling long videos [5, 6, 7], including excessive computational overhead, slow inference speeds, and performance degradation caused by redundant visual information.

Existing solutions primarily focus on token compression through saliency metrics derived from either LLM attention patterns [8, 9, 10] or visual encoder features [11, 12, 13]. While LLM attention-based methods selectively retain prompt-relevant tokens, their effectiveness heavily relies on layer-specific attention maps, introducing instability as shallow layers yield noisy signals while deeper layers negate computational benefits [14, 15]. Conversely, encoder feature-based approaches exploit intra-frame token sparsity through CLS token attention [16] or inter-patch correlations [17] but uniformly apply fixed compression ratios across frames, ignoring temporal dynamics critical for video understanding.

---

[*]Equal Contribution    [†] Corresponding Author

39th Conference on Neural Information Processing Systems (NeurIPS 2025).

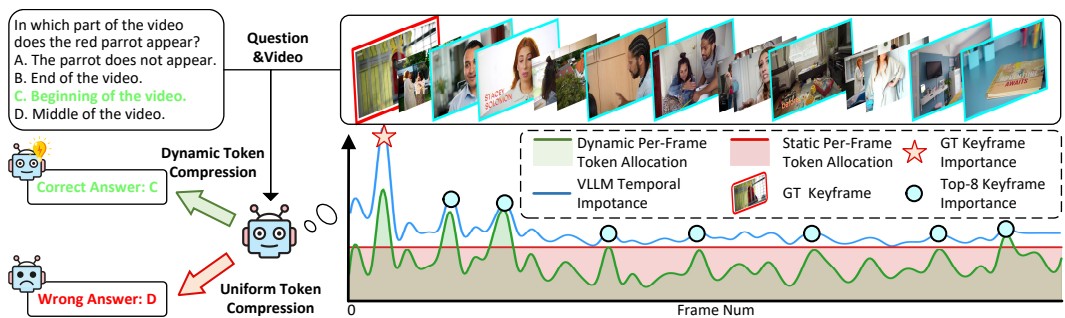

Figure 1: **Unveiling the keyframe prior in VLLMs.** LLaVA-OneVision's answers to video QA tasks are shown on the left. On the right, we plot the averaged attention from the final text token to visual tokens across all layers and within each frame. The top-8 frames by attention scores are arranged in time order, and the Ground Truth (GT) keyframes are highlighted in red. We observe that even when the model answers incorrectly, its attention still pinpoints the relevant frames, revealing a strong task-dependent keyframe prior.

**Motivation.** Unlike token compression for images, which applies a single ratio, video frames contain varying spatiotemporal information—some are critical for answering user queries, while others may hold less relevance. This inherent variation suggests that the shared compression ratios may be inappropriate for processing video frames. Consequently, a key question emerges: *How can we dynamically retain task-relevant information while filtering out the irrelevant components?*

An intuitive solution to this question is to perform the keyframe selection where frames deemed semantically relevant to user queries are prioritized while others are discarded, theoretically improving VLLMs' inference efficiency. However, existing works [18, 19] typically rely on pretrained vision-language models or auxiliary modules to perform frame selection before the feature encoding, introducing additional computational overhead that undermines the efficiency gains from frame reduction. Moreover, the binary selection paradigm proves suboptimal as it irrevocably discards potentially useful visual cues in unselected frames while retaining redundant information within chosen frames, creating an efficiency-utility trade-off that restricts video understanding. This observation suggests the need for *a more nuanced approach to spatial-temporal information compression for better preserving the task-relevant details within the constrained computational budget.*

**Our solution.** To address this challenge, in this work, we propose **Dy**namic **To**ken compression via LLM-guided **K**eyframe prior (**DyToK**), a simple yet effective training-free method that performs dynamic frame token compression for enhancing the efficiency of VLLMs. Instead of simply discarding or retaining frames outright, DyToK introduces a dynamic compression approach tailored for VLLMs. By leveraging the inherent attention mechanisms of VLLMs, DyToK selectively preserves the most critical tokens across frames, prioritizing those with higher importance. This ensures that keyframes retain more tokens, optimizing both efficiency and model performance. The modular design of DyToK ensures seamless compatibility with existing token pruning techniques, enabling plug-and-play integration without compromising efficiency.

Extensive evaluations on long-video benchmarks demonstrate DyToK's superiority over state-of-the-art methods. Under 20% token retention, our approach surpasses uncompressed baselines by 2.6% accuracy on LongVideoBench. At extreme compression ratios (10% retention), DyToK achieves a 24.0% performance gain over the competitors. To summarize, our contributions are as follows:

- We empirically reveal that the attention layers in VLLMs inherently encode query-conditioned keyframe priors that can be used for identifying the task-relevant information.

- We propose DyToK, a training-free compression paradigm that dynamically adjusts per-frame token retention ratios based on the model's intrinsic attention scores, achieving adaptive temporal compression while preserving task-critical semantics.

- DyToK demonstrates strong plug-and-play compatibility across different models and compression methods, attaining state-of-the-art efficiency-accuracy tradeoffs on three benchmarks specifically designed for long-video analysis.

## 2 Background and Motivation

In the following, we provide a concise overview of foundational concepts that underlie this study in Sec. 2.1, and highlight the key observations in Sec. 2.2, which offer valuable insights for motivating the proposed approach.

### 2.1 Preliminaries

**Architecture of VLLMs.** Modern VLLMs [1, 3, 4] are typically composed of three core components: a vision encoder, a modality projector, and a LLM backbone. The vision encoder, often pre-trained on large-scale image datasets (e.g., CLIP [16], SigLip [17]), processes each video frame into a sequence of visual tokens, with some variants incorporating video-specific pretraining [20, 21] for temporal modeling. The projector aligns these tokens with the LLM's textual embedding space, enabling cross-modal fusion. Finally, the LLM integrates the aligned visual and textual tokens to generate contextually relevant responses.

**Computational bottleneck analysis.** Though VLLMs have shown promising performance, the computational overhead impedes their practical applications. Specifically, the computational complexity of VLLMs is dominated by the self-attention mechanism and Feed-Forward Networks (FFNs) in transformer layers. For a model with $T$ transformer layers, the total Floating Point Operations (FLOPs) can be formulated as:

$$\text{Total FLOPs} = T \times \left(4nd^2 + 2n^2d + 2ndm\right), \tag{1}$$

where $n$ denotes the input sequence length, $d$ is the hidden dimension, and $m$ represents the FFN's intermediate size. In video tasks, $n$ is dominated by visual tokens $n_{\text{vis}}$, which often exceed textual tokens $n_{\text{sys}} + n_{\text{question}}$ by orders of magnitude through frame accumulation [12, 22]. Reducing $n_{\text{vis}}$ is thus crucial for improving inference efficiency.

**Efficient inference paradigms for VLLMs.** Recent advancements in efficient inference for VLLMs can be categorized into two paradigms: LLM attention-based token pruning [10, 13] and encoder feature-based token selection [23, 24]. The former (Fig. 2(a)) dynamically eliminates redundant visual tokens during LLM inference by leveraging cross-modal attention patterns from intermediate layers, yet suffers from unstable pruning decisions due to shallow-layer attention noise. The latter (Fig. 2(b)) statically selects tokens at the encoder output using feature correlations, but disregards temporal dependencies critical for video comprehension.

Notably, existing keyframe selection techniques [19, 18] primarily address issues arising from uniform input sampling in long videos, rather than optimizing for computational acceleration. Moreover, their binary frame selection approach risks discarding useful tokens of the discarded frames, potentially harming overall performance. Differently, in this study, we leverage a novel keyframe-aware compression strategy (Fig. 2(c)) that actively reduces inference latency for different frames, by which more tokens will be retained in the semantically important frames and fewer will be kept in those less relevant, thereby enhancing efficiency without compromising the coherence among the temporal semantics. A more comprehensive review of related works is provided in Appendix D.

### 2.2 Key Observations

Although prior compression methods [8, 13, 9, 10] have demonstrated promising results, key challenges persist. Most video understanding approaches [1, 2, 12, 9] directly adapt image-domain techniques, processing each frame independently while overlooking inter-frame correlations. However, in practice, video frames exhibit strong temporal dependencies—some contain query-critical information, while others contribute minimally. Therefore, to enable the dynamic frame-level token compression, we need to establish an accurate estimation of temporal importance across video frames.

We start by considering whether the attention map can serve as an indicator of the importance of frames. As shown in Fig. 1, we observe that the model consistently assigns peak attention scores to query-relevant keyframes, irrespective of answer correctness. This persistent pattern indicates an inherent bias toward semantically salient frames in the model's attention mechanism. A detailed analysis of keyframe prior in VLLMs is presented in Appendix B.

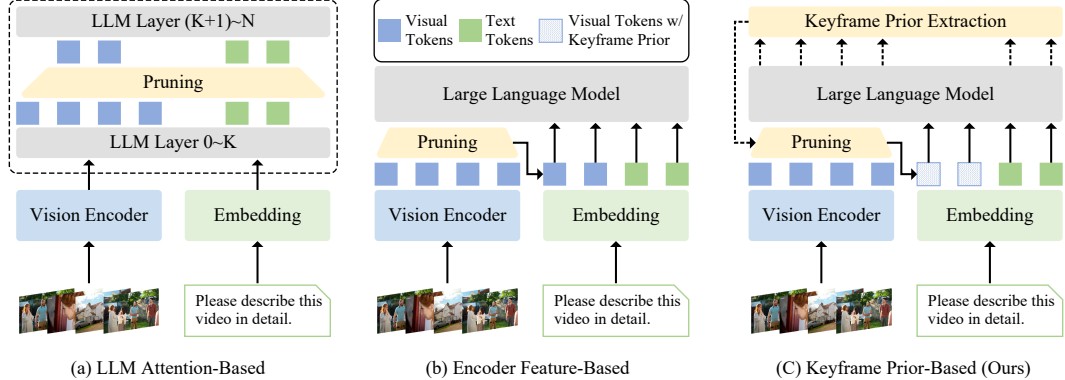

Figure 2: **Efficient inference methods for VLLMs.** (a) LLM attention-based methods perform token pruning during LLM inference by selecting visual tokens through cross-modal attention maps from specific layers, hence suffer from constrained pruning accuracy due to their reliance on noisy shallow-layer attention patterns. (b) Encoder feature-based methods prune tokens post-encoder using inter-patch feature correlations, but neglect temporal dynamics essential for video understanding. (c) Our approach uniquely exploits the keyframe priors embedded within LLMs to dynamically allocate frame-specific compression ratios, enabling plug-and-play enhancement of temporal perception capabilities in existing efficient VLLMs.

**Deep attention layers provide good keyframe priors.**   An intuitive approach is to aggregate attention patterns from all LLM layers for frame importance estimation. However, despite its effectiveness, this method necessitates full inference through the LLM, inevitably introducing computational inefficiency. Prior studies [25, 26] on transformer-based models show that shallow layers primarily capture local features, whereas deeper layers parse more abstract, high-level semantics.

To investigate this, we conduct a series of experiments. The results in Tab. 3 reveal that deeper layers exhibit more semantically meaningful and task-aware attention distributions, suggesting their potential as reliable keyframe indicators. However, directly using deep-layer attention to guide early layers within the same model introduces significant computational overhead—for instance, using the deepest layer to guide the first layer nearly doubles inference costs. To this end, a new challenge arises: *How can we leverage deep-layer attention keyframe priors for dynamic token compression while maintaining computational efficiency?*

## 3 Methodology

### 3.1 Overview

To address the above issues, we propose DyToK, a training-free framework that adaptively allocates per-frame compression ratios by leveraging the keyframe prior obtained from LLM. In Sec. 3.2, DyToK starts by Temporal Importance Estimation, where a lightweight assistant model computes frame-level importance scores, achieving a decent trade-off between efficiency and performance. Then, in Sec. 3.3, we present Dynamic Frame-Level Compression to distribute the overall token budget across frames according to these weights. This two-stage process yields a simple, effective, and general framework that integrates seamlessly with both encoder feature-based and LLM attention-based pruning paradigms. The overall framework of DyToK is demonstrated in Fig. 3.

### 3.2 Temporal Importance Estimation

Given a constrained budget for video token compression, it is necessary to allocate more tokens to critical frames. Thus, evaluating frame-level temporal importance scores is crucial to determine their relevance to the query, such that the token compression can be performed adaptively. To achieve this, we leverage cross-modal attention weights from the LLM to quantify frame importance.

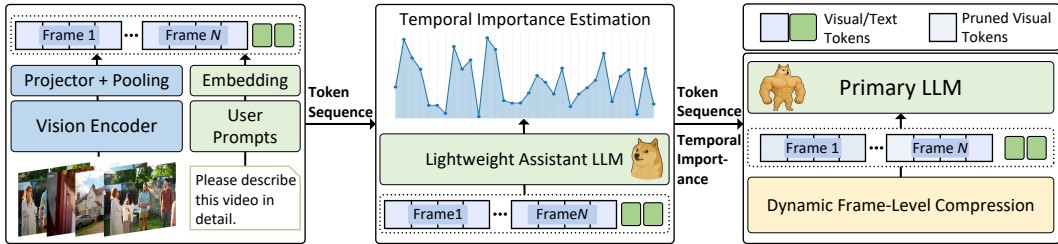

Figure 3: **Illustration of DyToK.** We adaptively compress video tokens through two synergistic components: (1) Temporal Importance Estimation leverages cross-modal attention from a lightweight assistant model to identify keyframes, followed by (2) Dynamic Frame-Level Compression that proportionally allocates token budgets to preserve salient content. This training-free paradigm achieves superior efficiency-accuracy tradeoffs by dynamically adjusting compression ratios per frame while maintaining compatibility with diverse token pruning methods.

Specifically, we compute attention scores between the last textual query token and visual tokens. Given an input video of $F$ frames, query and key vectors at attention layer $l$ are represented as $\mathbf{Q}_l \in \mathbb{R}^{1 \times D}$ and $\mathbf{K}_l \in \mathbb{R}^{V \times D}$, respectively. $V$ denotes the number of visual tokens, and $D$ is the embedding dimension per head. The importance $w_f$ of frame $f$ is then computed as:

$$w_f = \frac{1}{|\mathcal{L}|} \sum_{l \in \mathcal{L}} \mathrm{Softmax}\left(\frac{\mathbf{Q}_l \mathbf{K}_l^\top}{\sqrt{D}}\right), \tag{2}$$

where $\mathcal{L}$ is the set of layers considered for attention aggregation.

However, directly utilizing deep-layer attention to guide early layers within the same model introduces significant computational overhead due to the unavoidable re-computation of earlier layers. To address this, we propose leveraging an auxiliary lightweight assistant model derived from the same architectural family as the primary model to facilitate the generation of temporal resampling strategies. Experimental results in Tab. 4 reveal that this compact assistant model achieves nearly identical frame perception accuracy to the primary model despite being **14×** smaller, while occasionally demonstrating superior keyframe identification performance at certain compression ratios.

Furthermore, as discussed in Sec. 2.2, deeper layers tend to exhibit more semantically meaningful and task-relevant attention distributions, providing superior priors for frame importance evaluation. To leverage this property, we aggregate attention scores exclusively from the deep layers in the assistant model. To this end, the calibrated temporal importance score $\hat{w}_f$ is obtained as:

$$\hat{w}_f = \frac{1}{|\mathcal{L}'|} \sum_{l \in \mathcal{L}'} \mathrm{Softmax}\left(\frac{\mathbf{Q}_l \mathbf{K}_l^\top}{\sqrt{D}}\right), \tag{3}$$

where $\mathcal{L}'$ represents the selected subset of deep layers. The analysis of different layers, and the comparison between $w_f$ and $\hat{w}_f$ are provided in Sec. 4.2.

### 3.3 Dynamic Frame-Level Compression

Based on the temporal importance scores, in this section, we allocate token budgets proportionally to achieve frame-level dynamic compression, enhancing efficiency while preserving key information.

**Token budget allocation.** Specifically, the initial token assignment for each frame is computed as $a_f = \lfloor \hat{w}_f \times T_{total} \rfloor$, where $\hat{w}_f$ is the calibrated temporal importance score obtained in Eq. (3), $a_f$ denotes the preliminary token allocation for frame $f$, and $T_{\text{total}}$ defines the global token budget. Given $F$ frames within the input video, the set of remaining tokens $T_{\text{rem}}$ after the initial allocation is:

$$T_{\text{rem}} = T_{\text{total}} - \sum_{f=1}^{F} a_f. \tag{4}$$

To fairly distribute these remaining tokens, we calculate fractional remainders $r_f = (\hat{w}_f \times T_{\text{total}}) - a_f$ for each frame. Frames are then sorted by descending order of $r_f$, and the remaining tokens $T_{\text{rem}}$ are sequentially allocated to the top-ranked frames until exhausted. This ensures frames that were closest to receiving an additional token in the initial allocation get priority.

Then, to prevent excessive token allocation per frame, tokens exceeding the per-frame limit $T_{\text{max}}$ are reallocated to frames with available capacity according to their importance rankings. This yields the final token assignment as: $\hat{a}_f = \min(a_f, T_{\text{max}})$, with $\sum_{f=1}^{F} \hat{a}_f = T_{\text{total}}$ ensuring budget adherence.

**Dynamic token compression.** To perform token compression given the allocated budget, we define a modular compression function $\texttt{Compression}(x_f, a_f)$. Specifically, $x_f$ denotes the raw visual features (*e.g.*, patch tokens) for frame $f$, and $a_f$ represents the allocated token budget. This function can instantiate any compatible token pruning strategy, including both encoder feature-based approaches [11, 12, 13] and LLM attention-based token pruning methods [8, 9, 10]. The output of the function is the compressed token sequence $z_f$ for frame $f$:

$$z_f = \texttt{Compression}(x_f, a_f). \tag{5}$$

This design ensures that token reduction is tailored per frame based on its importance, while preserving compatibility with various compression backbones.

As summarized in Alg. 1, by adaptively allocating more token budget to the task-relevant critical frames, this strategy balances computational efficiency and performance, enabling dynamic adaptations to diverse video content characteristics. More implementation details and compatibility with different token compression methods are discussed in Sec. 4.1.

---

**Algorithm 1** Adaptive Token Allocation and Compression Based on Frame Weights

---

**Require:** Frame data $\{x_f\}_{f=1}^{F}$, importance weights $\{\hat{w}_f\}_{f=1}^{F}$, total budget $T_{\text{total}}$, per-frame token upper limit $T_{\text{max}}$
**Ensure:** Compressed tokens $\{z_f\}_{f=1}^{F}$
1:  Initialize allocation: $a_f \leftarrow \lfloor \hat{w}_f \times T_{\text{total}} \rfloor$
2:  Compute remaining tokens: $T_{\text{rem}} \leftarrow T_{\text{total}} - \sum_f a_f$
3:  Calculate fractional remainders: $r_f \leftarrow (\hat{w}_f \times T_{\text{total}}) - a_f$
4:  Sort frames by descending order of $r_f$
5:  **while** $T_{\text{rem}} > 0$ **do**
6:      **for** frame $f$ in sorted order **do**
7:         **if** $a_f < T_{\text{max}}$ **then**
8:            $a_f \leftarrow a_f + 1, T_{\text{rem}} \leftarrow T_{\text{rem}} - 1$
9:            **if** $T_{\text{rem}} = 0$ **then break**
10:         **end if**
11:      **end if**
12:    **end for**
13: **end while**
14: Redistribute excess tokens to frames below $T_{max}$ based on importance
15: **for** each frame $f \in \{1, ..., F\}$ **do**
16:    $z_f \leftarrow \texttt{Compression}(x_f, a_f)$        ▷ Apply dynamic per-frame compression
17: **end for**

---

## 4 Experiments

To evaluate DyToK, we conduct extensive experiments across multiple benchmarks and baseline VLLMs. The evaluation encompasses diverse video understanding tasks and token compression methods to assess both performance and generalizability. The subsequent section details the evaluation tasks, implementation settings, and key findings that align with our method.

### 4.1 Experimental Setup

We evaluate our method on three widely-used video understanding benchmarks: VideoMME [6], LongVideoBench [7], and MLVU [27], covering durations ranging from minutes to hours. Our

| Method | #Retained Tokens | VideoMME | | | | LongVideo Bench | MLVU | Average | |
|---|---|---|---|---|---|---|---|---|---|
| | | Short | Medium | Long | Overall | | | Score | % |
| Vanilla (TMLR) | 6272 | 70.0 | 56.7 | 48.8 | 58.5 | 56.6 | 47.1 | 54.1 | 100 |
| VisionZip (CVPR2025) | 4704 (↓ **25%**) | 68.9 | 56.2 | 48.2 | 57.8 | 55.4 | 45.0 | 52.7 | 97.4 |
| VisionZip† | | 71.4 | 55.2 | 49.0 | 58.6 | 56.0 | 46.0 | 53.5 | 98.9 |
| + DyToK | | 70.8 | 57.2 | 48.9 | 59.0 | 55.9 | 46.6 | 53.8 | 99.4 (+2.0) |
| DyCoke (CVPR2025) | | 70.9 | 56.4 | 48.8 | 58.7 | 55.3 | 47.5 | 53.8 | 99.4 |
| + DyToK | | 71.9 | 56.0 | 49.1 | 59.0 | 56.5 | 47.7 | 54.4 | 100.6 (+1.2) |
| VisionZip (CVPR2025) | 3136 (↓ **50%**) | 66.8 | 56.6 | 48.2 | 57.2 | 53.8 | 43.7 | 51.6 | 95.4 |
| VisionZip† | | 70.3 | 56.8 | 49.2 | 58.8 | 56.0 | 45.5 | 53.4 | 98.7 |
| + DyToK | | 71.9 | 56.1 | 49.3 | 59.1 | 56.4 | 46.2 | 53.9 | 99.6 (+4.2) |
| DyCoke (CVPR2025) | | 70.0 | 55.8 | 49.3 | 58.4 | 55.4 | 46.6 | 53.5 | 98.9 |
| + DyToK | | 70.2 | 55.9 | 48.9 | 58.3 | 57.1 | 47.5 | 54.3 | 100.4 (+1.5) |
| VisionZip (CVPR2025) | 1568 (↓ **75%**) | 62.2 | 53.0 | 47.4 | 54.2 | 51.4 | 41.2 | 48.9 | 90.4 |
| VisionZip† | | 68.8 | 57.3 | 48.1 | 58.1 | 55.8 | 44.8 | 52.9 | 97.8 |
| + DyToK | | 69.2 | 56.8 | 48.9 | 58.3 | 55.4 | 46.3 | 53.3 | 98.5 (+8.1) |
| DyCoke (CVPR2025) | | 68.2 | 56.1 | 47.7 | 57.3 | 54.6 | 43.5 | 51.8 | 95.7 |
| + DyToK | | 69.6 | 54.7 | 47.1 | 57.1 | 55.2 | 45.1 | 52.5 | 97.0 (+1.3) |
| VisionZip (CVPR2025) | 1120 (↓ **80%**) | 60.2 | 51.0 | 45.7 | 52.3 | 47.7 | 36.5 | 45.5 | 84.1 |
| VisionZip† | | 67.2 | 55.9 | 50.1 | 57.7 | 54.8 | 42.2 | 51.6 | 95.4 |
| + DyToK | | 69.2 | 56.4 | 47.8 | 57.8 | 56.0 | 45.2 | 53.0 | 98.0 (+13.9) |
| DyCoke (CVPR2025) | | 63.4 | 55.0 | 47.7 | 55.4 | 52.6 | 45.3 | 51.1 | 94.5 |
| + DyToK | | 65.1 | 54.2 | 47.3 | 55.6 | 53.2 | 44.6 | 51.1 | 94.5 |
| VisionZip (CVPR2025) | 448 (↓ **90%**) | 47.0 | 44.2 | 42.2 | 44.5 | 41.4 | 29.8 | 38.6 | 71.3 |
| VisionZip† | | 54.9 | 49.7 | 44.9 | 49.8 | 47.8 | 37.6 | 45.1 | 83.4 |
| + DyToK | | 61.0 | 51.1 | 47.6 | 53.2 | 50.4 | 42.8 | 48.8 | 90.2 (+18.9) |
| DyCoke (CVPR2025) | | 64.0 | 51.4 | 45.9 | 53.8 | 50.4 | 40.9 | 48.4 | 89.5 |
| + DyToK | | 62.3 | 51.6 | 46.6 | 53.5 | 50.2 | 43.6 | 49.1 | 90.8 (+1.3) |

Table 1: **Performance of DyToK integrated into encoder feature-based methods.** The vanilla setup processes 32 frames with 196 tokens each. The final column reports the average score across benchmarks and the accuracy relative to the unpruned baseline. VisionZip† denotes our pooling-compatible variant, and DyCoke here applies pruning exclusively at the encoder side.

evaluation follows the standard settings of LMMs-Eval [28]. Due to space limitations, we present only key results for LLaVA-OneVision [1] in the main text. To validate generalizability across different VLLM inference acceleration paradigms, we integrate DyToK into three state-of-the-art methods: FastV [8] (LLM attention-based), VisionZip [12] (encoder feature-based), and DyCoke [13] (hybrid pruning). For fair comparison, we align computational budgets to ensure equivalent FLOPs across methods (Appendix G.2). Comprehensive results are provided in the Appendix, including full experiments on 32-frame LLaVA-OneVision (Appendix A.1), evaluations of Qwen2.5-VL (Appendix A.2), analyses with extended video lengths (Appendix A.3), studies on broader model sizes (Appendix A.4), efficiency analysis (Appendix C), and implementation details (Appendix G).

### 4.2 Main Results

**Effectiveness on encoder feature-based methods.** We first evaluate DyToK on encoder feature-based VisionZip and DyCoke (encoder), which performs static token selection using encoder features. Notably, since VisionZip originally discards the spatial information of retained visual tokens, it is incompatible with 2D pooling methods, which are widespread in today's VLLMs. To address this limitation, we adapt VisionZip by deferring pruning until completely passing through the projection and pooling modules. Specifically, inter-patch correlations among pooled tokens serve as the pruning metric. We denote our improved variant as VisionZip†.

We report the average score on all the benchmarks and also in percentage format for comparative analysis, with the vanilla model's accuracy serving as the 100% upper limit to comprehensively assess the performance. As shown in Tab. 1, DyToK improves VisionZip by 4.2% average accuracy across

| Method | #Retained Tokens | VideoMME | | | | LongVideo Bench | MLVU | Average | |
|---|---|---|---|---|---|---|---|---|---|
| | | Short | Medium | Long | Overall | | | Score | % |
| Vanilla (TMLR) | 6272 | 70.0 | 56.7 | 48.8 | 58.5 | 56.6 | 47.1 | 54.1 | 100.0 |
| FastV (ECCV2024) | | 69.7 | 55.7 | 47.6 | 57.6 | 57.1 | 46.5 | 53.7 | 99.3 |
| + DyToK | 4704 | 70.4 | 56.7 | 48.2 | 58.4 | 56.8 | 46.8 | 54.0 | 99.8 (+0.5) |
| DyCoke (CVPR2025) | (↓ 25%) | 70.9 | 56.4 | 48.8 | 58.7 | 55.3 | 47.5 | 53.8 | 99.4 |
| + DyToK | | 70.9 | 55.9 | 49.0 | 58.6 | 55.9 | 47.5 | 54.0 | 99.8 (+0.4) |
| FastV (ECCV2024) | | 69.1 | 55.4 | 47.1 | 57.2 | 57.1 | 44.7 | 53 | 98.0 |
| + DyToK | 3136 | 71.1 | 55.6 | 48.8 | 58.5 | 57.2 | 46.3 | 54.0 | 99.8 (+1.8) |
| DyCoke (CVPR2025) | ↓ 50% | 70.0 | 55.8 | 49.3 | 58.4 | 55.4 | 46.6 | 53.5 | 98.9 |
| + DyToK | | 71.0 | 56.1 | 49.2 | 58.5 | 56.0 | 46.9 | 53.8 | 99.4 (+0.5) |
| FastV (ECCV2024) | | 66.2 | 54.7 | 47.1 | 56.0 | 56.6 | 43.7 | 52.1 | 96.3 |
| + DyToK | 1568 | 67.7 | 54.0 | 47.1 | 56.3 | 55.7 | 47.8 | 53.3 | 98.5 (+2.2) |
| DyCoke (CVPR2025) | (↓ 75%) | 68.2 | 56.1 | 47.7 | 57.3 | 54.6 | 43.5 | 51.8 | 95.7 |
| + DyToK | | 67.1 | 55.8 | 48.3 | 57.1 | 54.8 | 45.4 | 52.4 | 96.9 (+1.2) |
| FastV (ECCV2024) | | 58.0 | 51.7 | 43.6 | 51.1 | 51.2 | 38.3 | 46.9 | 86.7 |
| + DyToK | 896 | 64.6 | 54.2 | 45.8 | 54.8 | 52.6 | 43.2 | 50.2 | 92.8 (+6.1) |
| DyCoke (CVPR2025) | (↓ 85%) | 65.1 | 53.6 | 45.3 | 54.7 | 52.2 | 42.5 | 49.8 | 92.1 |
| + DyToK | | 66.1 | 52.8 | 46.3 | 55.1 | 52.4 | 42.9 | 50.1 | 92.6 (+0.5) |
| DyCoke (CVPR2025) | 448 | 64.0 | 51.4 | 45.9 | 53.8 | 50.4 | 40.9 | 48.4 | 89.5 |
| + DyToK | (↓ 90%) | 64.9 | 52.3 | 45.4 | 54.2 | 50.4 | 41.2 | 48.6 | 89.8 (+0.3) |

Table 2: **Performance of DyToK integrated into LLM attention-based methods.** The evaluation follows the same setup as above. DyCoke here only applies pruning at the LLM side.

benchmarks at 50% compression, demonstrating that our dynamic keyframe-aware compression complements static feature-based approaches. The performance gap widens as compression ratios increase (18.9% improvement at 90% compression), confirming that temporal importance awareness becomes critical under aggressive pruning.

**Effectiveness on LLM attention-based methods.** To enhance the compatibility of the proposed method with a broader range of existing pruning techniques, we also evaluate DyToK on LLM attention-based Methods, such as FastV and DyCoke (LLM). Following the same experimental setup, we report both the average score and the percentage relative to the unpruned LLaVA-OneVision baseline. As shown in Tab. 2, DyToK improves performance across both pruning rates. At an 85% compression rate, DyToK improves FastV by 6.1% in accuracy across benchmarks, further demonstrating the effectiveness and robustness of our method. Notably, at a 90% pruning rate, our token budget allocation (Appendix G.2) reduces FastV's retained visual tokens to zero and leaves DyToK with no tokens to reallocate, so results for this case are omitted.

### 4.3 Ablation Studies

**Effect of layer-level attention location on keyframe prior.** As hypothesized in Sec. 2.2, we conduct a detailed analysis to further illustrate this core finding. To evaluate the impact of different layers' attention patterns on keyframe prior, we fix the retention ratio at 20%—a moderate setting that effectively reduces redundancy without being overly aggressive. As shown in Tab. 3, we can clearly observe that deeper layers provide significantly better keyframe priors compared to shallower ones, with layers 20 and 23 achieving the best performance. The observed trend aligns with our hypothesis: model performance improves as deeper layers are used for keyframe prior extraction, which further proves the effectiveness of our method. To ensure generalizability without manual tuning, we uniformly average the attention scores from the last one-third of layers for all models in practical implementations. Please refer to Appendix A.6 for further details.

**Effect of the model size on keyframe prior.** As discussed in Sec. 2.2, using the full-size model to extract attention maps for token pruning is computationally expensive. We therefore evaluate whether a smaller model in the same VLLM family can assist pruning. We compare the pruning performance of the standard 7B model with that of its smaller counterpart across several video understanding tasks. Tab. 4 shows that the smaller model achieves comparable performance with significantly lower

| Layer | VideoMME | | | | LongVideo Bench | MLVU | Average | |
|---|---|---|---|---|---|---|---|---|
| | Short | Medium | Long | Overall | | | Score | % |
| Vanilla (TMLR) | 70.0 | 56.7 | 48.8 | 58.5 | 56.6 | 47.1 | 54.1 | 100.0 |
| 0 | 62.6 | 53.1 | 46.4 | 54.0 | 52.0 | 40.8 | 48.9 | 90.5 |
| 4 | 63.6 | 53.3 | 49.0 | 55.3 | 53.5 | 42.3 | 50.4 | 93.1 |
| 8 | 64.1 | 53.4 | 47.3 | 55.0 | 53.4 | 41.5 | 50.0 | 92.3 |
| 12 | 65.7 | 54.9 | 47.8 | 56.1 | 53.3 | 43.3 | 50.9 | 94.1 |
| 16 | 66.2 | 55.3 | 48.2 | 56.6 | 53.4 | 44.9 | 51.6 | 95.4 |
| 20 | 68.1 | 56.6 | 47.8 | 57.5 | 55.5 | 46.3 | 53.1 | 98.2 |
| 23 | 66.3 | 56.3 | 48.7 | 57.1 | 53.8 | 45.1 | **52.0** | **96.1** |

Table 3: **Performance of DyToK under different layer configurations.** We conduct experiments using a retention ratio of 20%, with the number of frames set to 32. In comparison, the baseline LLaVA-OneVision retains its original configuration without any pruning or modification. The best result is highlighted in red, while the second-best is shown in **bold**.

| Method | #Retained Tokens | VideoMME | | | | LongVideo Bench | MLVU | Average | |
|---|---|---|---|---|---|---|---|---|---|
| | | Short | Medium | Long | Overall | | | Score | % |
| Vanilla (TMLR) | 6272 | 70.0 | 56.7 | 48.8 | 58.5 | 56.6 | 47.1 | 54.1 | 100.0 |
| Base | 4704 | 71.9 | 56.8 | 48.8 | 59.1 | 55.7 | 47.5 | 54.1 | 100.0 |
| Tiny | (↓ **25%**) | 70.8 | 57.2 | 48.9 | 59.0 | 55.9 | 46.6 | 53.8 | 99.4 (−0.6) |
| Base | 3136 | 71.8 | 57.2 | 49.0 | 59.3 | 55.9 | 46.0 | 53.7 | 99.3 |
| Tiny | ↓ **50%** | 71.9 | 56.1 | 49.3 | 59.1 | 56.4 | 46.2 | 53.9 | 99.6 (+0.3) |
| Base | 1568 | 70.4 | 57.1 | 48.1 | 58.6 | 57.2 | 46.6 | 54.1 | 100.0 |
| Tiny | (↓ **75%**) | 69.2 | 56.8 | 48.9 | 58.3 | 55.4 | 46.3 | 53.3 | 98.5 (−1.5) |
| Base | 1120 | 70.3 | 56.4 | 48.1 | 58.3 | 55.8 | 46.5 | 53.5 | 98.9 |
| Tiny | (↓ **80%**) | 69.2 | 56.4 | 47.8 | 57.8 | 56.0 | 45.2 | 53.0 | 98.0 (−0.9) |
| Base | 448 | 63.9 | 52.7 | 47.3 | 54.6 | 52.7 | 41.5 | 49.6 | 91.7 |
| Tiny | (↓ **90%**) | 61.0 | 51.1 | 47.6 | 53.2 | 50.4 | 42.8 | 48.8 | 90.2 (−1.5) |

Table 4: **Performance of DyToK with different LLM sizes for keyframe prior.** The *Base* setting uses LLaVA-OneVision-7B to provide the keyframe prior, while the *Tiny* setting employs LLaVA-OneVision-0.5B for the same purpose. All experiments are conducted with 32 input frames.

computational cost ($14\times$) while incurring minimal performance degradation. More ablations on token allocation upper limit (Appendix A.7), importance estimation strategies (Appendix A.8), and textual token selection (Appendix A.9) are detailed in the Appendix.

## 5 Concluding Remarks

**Summary.** In this paper, we analyze attention maps in VLLMs and observe a strong correlation between attention scores and keyframes, along with the insight that deeper layers provide more effective priors for keyframe selection. Building on these findings, we propose DyToK, a simple yet effective method that leverages the inherent priors of LLMs to reduce vision tokens while preserving video understanding capabilities. DyToK is a plug-and-play method compatible with both attention- and encoder-based approaches, demonstrating broad applicability across different architectures.

**Limitation and future work.** Although employing a lightweight assistant model to approximate the original model for keyframe prior generation improves performance while ensuring efficiency, this work has not yet to propose a better method to avoid introducing additional models. Future efforts will focus on further optimization to address this issue.

**Acknowledgement.** This work was supported by Guangdong Basic and Applied Basic Research Foundation (2025A1515011546), by Shenzhen Science and Technology Innovation Program (JCYJ20240813105901003, KJZD20240903102901003), and by Science and Technology Project of Shenzhen (GXWD-202208111170603002).

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

# Supplementary Material

## Contents

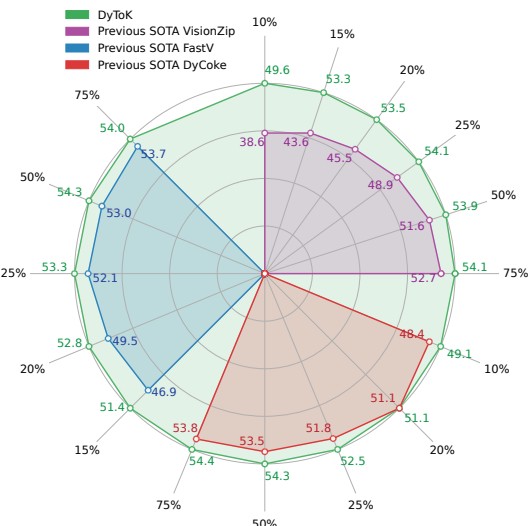

Figure 4: **Performance gains of DyToK under various retention ratios.** Performance comparison of SOTA acceleration methods with and without DyToK on LLaVA-OneVision under 32-frame input. Experiments conducted on VideoMME, LongVideoBench, and MLVU across varying retention ratios show that integrating DyToK consistently improves accuracy, demonstrating its effectiveness in enhancing long-video understanding. The scores presented in the figure represent average performance across the three benchmarks. For detailed results, please refer to Tab. 5 and Tab. 6

## A  More Experimental Results

### A.1  Full Experiments on 32-Frame LLaVA-OneVision

In the Tab. 1 and Tab. 2 of the main text, we evaluate the effectiveness of DyToK using a lightweight assistant model to generate keyframe priors for token pruning under a 32-frame setting with LLaVA-OneVision. To further validate the generalizability and efficacy of DyToK, this section expands upon those findings by employing the primary model's attention scores directly as temporal importance indicators to dynamically allocate token budgets. Experiments are conducted across three widely recognized benchmarks: VideoMME, LongVideoBench, and MLVU. Results corresponding to identical experimental conditions in the main text are aligned for consistency.

As demonstrated in Tab. 5 and Tab. 6, DyToK consistently enhances performance across different VLLM inference acceleration paradigms, specifically encoder feature-based and LLM attention-based methods. Under relatively moderate token reduction ratios (25%-50%), DyToK improves upon baseline pruning methods by effectively preserving keyframe tokens, resulting in notable gains of up to 4.2% (VisionZip, 50% token reduction) and up to 2.4% (FastV, 50% token reduction) in average scores. Notably, at more aggressive compression settings (75%-90% token reductions), the advantage of DyToK becomes even more pronounced, delivering substantial accuracy improvements—up to 20.4% for VisionZip at 90% reduction and 9.6% at 75% reduction, clearly highlighting DyToK's capability to significantly mitigate accuracy degradation under extreme token pruning scenarios.

### A.2  Experiments on Qwen2.5-VL

To further demonstrate the generalization capability of our method, we conduct experiments on the recently released Qwen2.5-VL [3] model, which has gained prominence in the current VLM community. This model introduces architectural components not aligned with previous pruning methods, including sliding window attention and 3D convolutions—both of which are rarely present in earlier models. As a result, we adapt the logic of existing state-of-the-art pruning methods to suit this architecture. The experiments are conducted using uniform sampling of 32 frames with the Qwen2.5-VL 7B model. Detailed results are presented in Tab. 7.

Notably, across all retention ratios when used with FastV, our method achieves a substantial performance gain—exceeding 10 points in most settings. Remarkably, it even outperforms the vanilla

| Method | VideoMME | | | | LongVideo Bench | MLVU | Average | |
|---|---|---|---|---|---|---|---|---|
| | Short | Medium | Long | Overall | | | Score | % |
| *Upper Bound, 6272 tokens* | | | | | | | | |
| Vanilla (TMLR) | 70.0 | 56.7 | 48.8 | 58.5 | 56.6 | 47.1 | 54.1 | 100 |
| *Retain 4704 tokens (↓ 25%)* | | | | | | | | |
| VisionZip (CVPR2025) | 68.9 | 56.2 | 48.2 | 57.8 | 55.4 | 45.0 | 52.7 | 97.4 |
| VisionZip[†] | 71.4 | 55.2 | 49.0 | 58.6 | 56.0 | 46.0 | 53.5 | 98.9 |
| + DyToK | 70.8 | 57.2 | 48.9 | 59.0 | 55.9 | 46.6 | 53.8 | 99.4 (+2.0) |
| + DyToK (7B) | 71.9 | 56.8 | 48.8 | 59.1 | 55.7 | 47.5 | 54.1 | 100 (+2.6) |
| DyCoke (CVPR2025) | 70.9 | 56.4 | 48.8 | 58.7 | 55.3 | 47.5 | 53.8 | 99.4 |
| + DyToK | 71.9 | 56.0 | 49.1 | 59.0 | 56.5 | 47.7 | 54.4 | 100.6 (+1.2) |
| + DyToK (7B) | 72.0 | 55.9 | 49.0 | 59.0 | 56.3 | 47.6 | 54.3 | 100.4 (+1.0) |
| *Retain 3136 tokens (↓ 50%)* | | | | | | | | |
| VisionZip (CVPR2025) | 66.8 | 56.6 | 48.2 | 57.2 | 53.8 | 43.7 | 51.6 | 95.4 |
| VisionZip[†] | 70.3 | 56.8 | 49.2 | 58.8 | 56.0 | 45.5 | 53.4 | 98.7 |
| + DyToK | 71.9 | 56.1 | 49.3 | 59.1 | 56.4 | 46.2 | 53.9 | 99.6 (+4.2) |
| + DyToK (7B) | 71.8 | 57.2 | 49.0 | 59.3 | 55.9 | 46.0 | 53.7 | 99.3 (+3.9) |
| DyCoke (CVPR2025) | 70.0 | 55.8 | 49.3 | 58.4 | 55.4 | 46.59 | 53.5 | 98.9 |
| + DyToK | 70.2 | 55.9 | 48.9 | 58.3 | 57.1 | 47.5 | 54.3 | 100.4 (+1.5) |
| + DyToK(7B) | 71.0 | 56.7 | 49.2 | 59.0 | 56.6 | 47.1 | 54.2 | 100.2 (+1.3) |
| *Retain 1568 tokens (↓ 75%)* | | | | | | | | |
| VisionZip (CVPR2025) | 62.2 | 53.0 | 47.4 | 54.2 | 51.4 | 41.2 | 48.9 | 90.4 |
| VisionZip[†] | 68.8 | 57.3 | 48.1 | 58.1 | 55.8 | 44.8 | 52.9 | 97.8 |
| + DyToK | 69.2 | 56.8 | 48.9 | 58.3 | 55.4 | 46.3 | 53.3 | 98.5 (+8.1) |
| + DyToK (7B) | 70.4 | 57.1 | 48.1 | 58.6 | 57.2 | 46.6 | 54.1 | 100 (+9.6) |
| DyCoke (CVPR2025) | 68.2 | 56.1 | 47.7 | 57.3 | 54.6 | 43.5 | 51.8 | 95.7 |
| + DyToK | 69.6 | 54.7 | 47.1 | 57.1 | 55.2 | 45.1 | 52.5 | 97.0 (+1.3) |
| + DyToK(7B) | 68.0 | 54.9 | 47.2 | 56.7 | 53.7 | 45.6 | 52.0 | 96.1 (+0.4) |
| *Retain 1120 tokens (↓ 80%)* | | | | | | | | |
| VisionZip (CVPR2025) | 60.2 | 51.0 | 45.7 | 52.3 | 47.7 | 36.5 | 45.5 | 84.1 |
| VisionZip[†] | 67.2 | 55.9 | 50.1 | 57.7 | 54.8 | 42.2 | 51.6 | 95.4 |
| + DyToK | 69.2 | 56.4 | 47.8 | 57.8 | 56.0 | 45.2 | 53.0 | 98.0 (+13.9) |
| + DyToK (7B) | 70.3 | 56.4 | 48.1 | 58.3 | 55.8 | 46.5 | 53.5 | 98.9 (+14.8) |
| DyCoke (CVPR2025) | 63.4 | 55.0 | 47.7 | 55.4 | 52.6 | 45.3 | 51.1 | 94.5 |
| + DyToK | 65.1 | 54.2 | 47.3 | 55.6 | 53.2 | 44.6 | 51.1 | 94.5 (+0.0) |
| + DyToK (7B) | 66.6 | 53.4 | 46.7 | 55.6 | 52.4 | 44.9 | 51.0 | 94.3 (−0.2) |
| *Retain 448 tokens (↓ 90%)* | | | | | | | | |
| VisionZip (CVPR2025) | 47.0 | 44.2 | 42.2 | 44.5 | 41.4 | 29.8 | 38.6 | 71.3 |
| VisionZip[†] | 54.9 | 49.7 | 44.9 | 49.8 | 47.8 | 37.6 | 45.1 | 83.4 |
| + DyToK | 61.0 | 51.1 | 47.6 | 53.2 | 50.4 | 42.8 | 48.8 | 90.2 (+18.9) |
| + DyToK (7B) | 63.9 | 52.7 | 47.3 | 54.6 | 52.7 | 41.5 | 49.6 | 91.7 (+20.4) |
| DyCoke (CVPR2025) | 64.0 | 51.4 | 45.9 | 53.8 | 50.4 | 40.9 | 48.4 | 89.5 |
| + DyToK | 62.3 | 51.6 | 46.6 | 53.5 | 50.2 | 43.6 | 49.1 | 90.8 (+1.3) |
| + DyToK (7B) | 61.4 | 52.1 | 46.1 | 53.2 | 50.6 | 41.3 | 48.4 | 89.5 (+0.0) |

Table 5: **Full experiments of DyToK integrated into encoder feature-based methods on 32-frames LLaVA-OneVision.** The vanilla setup processes 32 frames with 196 tokens each. The final column reports the average score across benchmarks and the accuracy relative to the unpruned baseline. VisionZip[†] denotes our pooling-compatible variant, and DyCoke here applies dynamic per-frame compression ratio allocation exclusively at the encoder side.

| Method | VideoMME | | | | LongVideo Bench | MLVU | Average | |
|---|---|---|---|---|---|---|---|---|
| | Short | Medium | Long | Overall | | | Score | % |
| *Upper Bound, 6272 tokens* | | | | | | | | |
| Vanilla (TMLR) | 70.0 | 56.7 | 48.8 | 58.5 | 56.6 | 47.1 | 54.1 | 100 |
| *Retain 4704 tokens (↓ 25%)* | | | | | | | | |
| FastV (ECCV2024) | 69.7 | 55.7 | 47.6 | 57.6 | 57.1 | 46.5 | 53.7 | 99.3 |
| + DyToK | 70.4 | 56.7 | 48.2 | 58.4 | 56.8 | 46.8 | 54.0 | 99.8 (+0.5) |
| + DyToK (7B) | 70.1 | 55.9 | 48.4 | 58.1 | 57.6 | 46.4 | 54.0 | 99.8 (+0.5) |
| DyCoke (CVPR2025) | 70.9 | 56.4 | 48.8 | 58.7 | 55.3 | 47.5 | 53.8 | 99.4 |
| + DyToK | 70.9 | 55.9 | 49.0 | 58.6 | 55.9 | 47.5 | 54 | 99.8 (+0.4) |
| + DyToK (7B) | 70.9 | 55.7 | 48.6 | 58.4 | 55.7 | 47.7 | 53.9 | 99.6 (+0.2) |
| *Retain 3136 tokens (↓ 50%)* | | | | | | | | |
| FastV (ECCV2024) | 69.1 | 55.4 | 47.1 | 57.2 | 57.1 | 44.7 | 53 | 98.0 |
| + DyToK | 71.1 | 55.6 | 48.8 | 58.5 | 57.2 | 46.3 | 54 | 99.8 (+1.8) |
| + DyToK (7B) | 71.1 | 56.3 | 47.6 | 58.3 | 57.4 | 47.1 | 54.3 | 100.4 (+2.4) |
| DyCoke (CVPR2025) | 70.0 | 55.8 | 49.3 | 58.4 | 55.4 | 46.6 | 53.5 | 98.9 |
| + DyToK | 71.0 | 56.1 | 49.2 | 58.5 | 56.0 | 46.9 | 53.8 | 99.4 (+0.5) |
| + DyToK (7B) | 70.7 | 55.9 | 48.9 | 58.5 | 56.1 | 46.7 | 53.8 | 99.4 (+0.5) |
| *Retain 1568 tokens (↓ 75%)* | | | | | | | | |
| FastV (ECCV2024) | 66.2 | 54.7 | 47.1 | 56.0 | 56.6 | 43.7 | 52.1 | 96.3 |
| + DyToK | 67.7 | 54.0 | 47.1 | 56.3 | 55.7 | 47.8 | 53.3 | 98.5 (+2.2) |
| + DyToK (7B) | 69.4 | 56.1 | 47.2 | 57.6 | 56.0 | 45.8 | 53.1 | 98.2 (+1.9) |
| DyCoke (CVPR2025) | 68.2 | 56.1 | 47.7 | 57.3 | 54.6 | 43.5 | 51.8 | 95.7 |
| + DyToK | 67.1 | 55.8 | 48.3 | 57.1 | 54.8 | 45.4 | 52.4 | 96.9 (+1.2) |
| + DyToK (7B) | 68.6 | 55.2 | 49.0 | 57.6 | 55.2 | 46.2 | 53 | 98.0 (+1.1) |
| *Retain 896 tokens (↓ 85%)* | | | | | | | | |
| FastV (ECCV2024) | 58.0 | 51.7 | 43.6 | 51.1 | 51.2 | 38.3 | 46.9 | 86.7 |
| + DyToK | 64.6 | 54.2 | 45.8 | 54.8 | 52.6 | 43.2 | 50.2 | 92.8 (+6.1) |
| + DyToK (7B) | 67.1 | 55.1 | 46.0 | 56.1 | 54.0 | 44.0 | 51.4 | 95.0 (+8.3) |
| DyCoke (CVPR2025) | 65.1 | 53.6 | 45.3 | 54.7 | 52.2 | 42.5 | 49.8 | 92.1 |
| + DyToK | 66.1 | 52.8 | 46.3 | 55.1 | 52.4 | 42.9 | 50.1 | 92.6 (+0.5) |
| + DyToK (7B) | 65.4 | 52.7 | 46.3 | 54.8 | 52.7 | 41.9 | 49.8 | 92.1 (+0.0) |
| *Retain 448 tokens (↓ 90%)* | | | | | | | | |
| DyCoke (CVPR2025) | 64.0 | 51.4 | 45.9 | 53.8 | 50.4 | 40.9 | 48.4 | 89.5 |
| + DyToK | 64.9 | 52.3 | 45.4 | 54.2 | 50.41 | 41.2 | 48.6 | 89.8 (+0.3) |
| + DyToK (7B) | 64.3 | 51.9 | 45.1 | 53.8 | 51.3 | 42.5 | 49.2 | 90.9 (+1.4) |

Table 6: **Full experiments of DyToK integrated into LLM attention-based methods on 32-frames LLaVA-OneVision.** The evaluation follows the same setup as above. DyCoke here only applies dynamic per-frame compression ratio allocation at the LLM side.

| Method | VideoMME | LongVideoBench | MLVU | Average | |
|---|---|---|---|---|---|
| | | | | Score | % |
| *Upper Bound* | | | | | |
| Vanilla (TMLR) | 60.7 | 57.7 | 44.7 | 54.4 | 100 |
| *Retain 75% tokens* | | | | | |
| VisionZip† | 59.7 | 56.5 | 44.6 | 53.6 | 98.5 |
| + DyToK (7B) | 59.3 | 57.0 | 43.1 | 53.1 | 97.6 (−0.9) |
| FastV (ECCV2024) | 58.7 | 56.1 | 41.5 | 52.1 | 96.3 |
| + DyToK (7B) | 60.7 | 58.3 | 45.4 | 54.8 | 100.1 (+3.8) |
| *Retain 50% tokens* | | | | | |
| VisionZip† | 59.6 | 56.8 | 45.0 | 53.8 | 98.9 |
| + DyToK (7B) | 60.2 | 56.8 | 43.8 | 53.6 | 98.5 (−0.4) |
| FastV (ECCV2024) | 55.7 | 53.2 | 36.8 | 48.6 | 89.3 |
| + DyToK(7B) | 60.8 | 57.9 | 44.1 | 54.3 | 99.8 (+10.5) |
| *Retain 25% tokens* | | | | | |
| VisionZip† | 59.2 | 55.1 | 44.3 | 52.9 | 97.2 |
| + DyToK (7B) | 59.9 | 55.3 | 44.5 | 53.2 | 97.8 (+0.6) |
| FastV (ECCV2024) | 53.1 | 49.3 | 34.0 | 45.5 | 83.6 |
| + DyToK(7B) | 58.8 | 54.5 | 43.3 | 52.2 | 96.0 (+12.4) |
| *Retain 15% tokens* | | | | | |
| VisionZip† | 58.3 | 55.4 | 41.6 | 51.8 | 95.2 |
| + DyToK (7B) | 58.4 | 54.1 | 42.5 | 51.7 | 95.0 (−0.2) |
| FastV (ECCV2024) | 48.9 | 47.1 | 32.8 | 42.9 | 78.9 |
| + DyToK (7B) | 56.9 | 51.1 | 39.8 | 49.3 | 90.6 (+11.7) |
| *Retain 10% tokens* | | | | | |
| VisionZip† | 55.8 | 53.2 | 41.7 | 50.2 | 92.3 |
| + DyToK (7B) | 57.1 | 53.1 | 43.6 | 51.3 | 94.3 (+2.0) |
| FastV (ECCV2024) | 42.3 | 42.3 | 24.6 | 36.4 | 66.9 |
| + DyToK (7B) | 42.3 | 42.3 | 24.6 | 36.4 | 66.9 (+0.0) |

Table 7: **Performance of DyToK integrated into VisionZip and FastV on Qwen2.5-VL.** The vanilla setup processes 32 frames. The final column reports the average score across benchmarks and the accuracy relative to the unpruned baseline. VisionZip† denotes our pooling-compatible variant.

(non-pruned) setting at a 75% retention ratio by 0.1%, demonstrating the strong effectiveness of our approach. In the extreme case of a 10% retention ratio, our method combined with VisionZip also yields a 2% improvement, further confirming its robustness under aggressive compression.

## A.3 Experiments on Extended Video Length

To further verify DyToK's effectiveness on longer video sequences, we conduct supplementary experiments using a uniform sampling of 64 frames with LLaVA-OneVision, as presented in Tab. 8. This experiment complements the findings in the main text by evaluating DyToK's scalability and robustness when dealing with increased temporal context.

The experimental results demonstrate DyToK's consistent superiority across various token retention ratios. At moderate compression levels (25%-50%), DyToK notably enhances performance, achieving up to 7.0% improvement in average accuracy compared to baseline methods at a 50% token retention ratio. Remarkably, under more severe compression conditions (75%-90% token reduction), DyToK's advantage becomes even more pronounced. Specifically, at a retention ratio of just 10% of the original tokens (896 tokens), DyToK achieves substantial accuracy gains of up to 24.0% over baseline methods, highlighting its capability to preserve essential temporal information efficiently.

| Method | VideoMME | | | | LongVideo Bench | MLVU | Average | |
|---|---|---|---|---|---|---|---|---|
| | Short | Medium | Long | Overall | | | Score | % |
| *Upper Bound, 12544 tokens* | | | | | | | | |
| Vanilla (TMLR) | 70.8 | 56.9 | 48.7 | 58.8 | 57.7 | 50.0 | 55.5 | 100 |
| *Retain 9408 tokens (↓ 25%)* | | | | | | | | |
| VisionZip (CVPR 2025) | 70.3 | 55.7 | 50.1 | 58.7 | 56.0 | 43.9 | 52.9 | 95.3 |
| VisionZip† | 71.1 | 56.8 | 48.9 | 58.9 | 58.0 | 49.6 | 55.5 | 100 |
| + DyTok | 71.0 | 56.6 | 49.7 | 59.1 | 58.2 | 48.6 | 55.3 | 99.6 (+4.3) |
| + DyTok (7B) | 70.8 | 57.2 | 48.4 | 58.5 | 58.3 | 49.1 | 55.3 | 99.6 (+4.3) |
| *Retain 6272 tokens (↓ 50%)* | | | | | | | | |
| VisionZip (CVPR 2025) | 69.0 | 57.1 | 49.1 | 58.4 | 55.2 | 42.3 | 52.0 | 93.7 |
| VisionZip† | 72.4 | 56.0 | 48.4 | 59.0 | 57.7 | 48.7 | 55.1 | 99.3 |
| + DyTok | 72.1 | 57.8 | 49.2 | 59.7 | 58.5 | 48.1 | 55.4 | 99.8 (+6.1) |
| + DyTok (7B) | 72.2 | 58.1 | 49.0 | 59.8 | 58.3 | 49.7 | 55.9 | 100.7 (+7.0) |
| *Retain 3136 tokens (↓ 75%)* | | | | | | | | |
| VisionZip (CVPR 2025) | 65.6 | 53.4 | 48.2 | 55.7 | 51.5 | 42.3 | 49.8 | 89.7 |
| VisionZip† | 70.3 | 55.8 | 48.1 | 58.1 | 56.6 | 47.4 | 54.0 | 97.3 |
| + DyTok | 71.6 | 57.4 | 49.1 | 59.4 | 57.8 | 46.5 | 54.6 | 98.4 (+8.7) |
| + DyTok (7B) | 72.7 | 57.7 | 48.1 | 59.5 | 58.4 | 47.4 | 55.1 | 99.3 (+9.6) |
| *Retain 2240 tokens (↓ 80%)* | | | | | | | | |
| VisionZip (CVPR 2025) | 61.6 | 52.9 | 46.1 | 53.5 | 48.5 | 37.1 | 46.4 | 83.6 |
| VisionZip† | 70.1 | 56.4 | 48.0 | 58.2 | 55.5 | 46.8 | 53.5 | 96.4 |
| + DyTok | 70.1 | 56.3 | 48.7 | 58.4 | 56.1 | 46.1 | 53.5 | 96.4 (+12.8) |
| + DyTok (7B) | 71.6 | 57.1 | 47.8 | 58.8 | 59.2 | 47.7 | 55.2 | 99.5 (+15.9) |
| *Retain 1792 tokens (↓ 85%)* | | | | | | | | |
| VisionZip (CVPR 2025) | 58.1 | 50.6 | 44.3 | 51.0 | 46.7 | 34.0 | 43.9 | 79.1 |
| VisionZip† | 67.4 | 54.8 | 47.1 | 56.4 | 54.5 | 42.9 | 51.3 | 92.4 |
| + DyTok | 69.0 | 55.8 | 48.1 | 57.6 | 55.3 | 45.0 | 52.6 | 94.8 (+15.7) |
| + DyTok (7B) | 70.6 | 56.9 | 47.9 | 58.4 | 57.7 | 46.5 | 54.2 | 97.7 (+18.6) |
| *Retain 896 tokens (↓ 90%)* | | | | | | | | |
| VisionZip (CVPR 2025) | 47.8 | 45.3 | 41.9 | 45.0 | 42.1 | 29.0 | 38.7 | 69.7 |
| VisionZip† | 56.7 | 52.1 | 45.9 | 51.6 | 47.9 | 36.8 | 45.4 | 81.8 |
| + DyTok | 64.0 | 52.9 | 47.2 | 54.7 | 53.6 | 42.6 | 50.3 | 90.6 (+20.9) |
| + DyTok (7B) | 65.9 | 55.8 | 46.3 | 56.0 | 55.1 | 45.0 | 52.0 | 93.7 (+24.0) |

Table 8: **Performance of DyToK on 64-frames LLaVA-OneVision.** The vanilla setup processes 64 frames with 196 tokens each. The final column reports the average score across benchmarks and the accuracy relative to the unpruned baseline. VisionZip† denotes our pooling-compatible variant. DyTok(7B) indicates that token pruning guided by 7B model.

| Method | 100% | 25% | 20% | 15% |
|--------|------|-----|-----|-----|
| Vanilla | 41.4 | N/A | N/A | N/A |
| FastV | N/A | 23.6 | 22.3 | 17.1 |
| DyToK (3B guides 32B) | N/A | 35.5 | **35.4** | 29.9 |
| DyToK (7B guides 32B) | N/A | **36.3** | 34.9 | **33.9** |

Table 9: **Experiments on broader model size.** We conduct experiments utilizing Qwen2.5-VL (3B, 7B, and 32B) on the MLVU benchmark. Specifically, models are evaluated using 32 input frames, with retention ratios ranging from 15% to 25%, based on the FastV pruning method.

Overall, these findings confirm DyToK's effectiveness in significantly mitigating performance degradation under extreme compression scenarios, reinforcing its general applicability and robustness for longer video sequences.

## A.4 Experiments on Broader Model Size

Beyond the 0.5B guiding 7B experiments using LLaVA-OneVision already presented in the main text, we conduct additional extensive experiments using Qwen2.5-VL on the MLVU benchmark, which covers a wider range of commonly used model sizes (3B, 7B, and 32B), as shown in Tab. 9. For the experiments in this section, we adopt a 4-bit NormalFloat (NF4) quantization configuration with double quantization and bfloat16 compute, and enable FlashAttention-2 during inference. The use of these heterogeneous quantization regimes is a pragmatic choice dictated by hardware-resource constraints, primarily regarding device memory capacity and memory bandwidth/throughput characteristics, on our evaluation platforms.

These experimental results provide two further insights. First, video models inherently possess keyframe priors regardless of model size. Second, such priors are not constrained by model scale, as even lightweight models can exhibit keyframe perception capabilities comparable to those of larger models. Notably, at a 20% retention ratio, the 3B Qwen2.5-VL assistant model surpasses the 7B variant by a substantial margin, further validating the feasibility of using a lightweight assistant model to provide keyframe priors in DyToK.

## A.5 Experiments on General Descriptive Queries

To assess DyToK's ability to select keyframes for general descriptive queries (i.e., the query does not explicitly point to specific visual content), we conduct captioning experiments on VideoChatGPT [29] using LLaVA-OneVision (7B), which includes prompts like "What is happening in the video?" Following Dynamic-VLM [30], we use GPT-3.5-Turbo-0613 for quantitative scoring, with the results summarized in Tab. 10.

We attribute DyToK's advantage to its effectiveness in identifying both "semantic keyframes" (relevant to query semantics) and "content keyframes" (capturing key scene or event transitions), making it well-suited for general descriptive queries. This distinguishes DyToK from CLIP-based methods, which rely solely on semantic matching. To support our claim, we replace DyToK's cross-modal attention-based frame weight estimation with CLIP-based semantic similarity. Implementation details follow AKS [19]. Results remain consistent, as shown in Tab. 11.

## A.6 Ablation on Keyframe Prior Layer Selection

In the main text (Tab. 3), we partially discuss the effectiveness of extracting keyframe priors from different layers of LLM. To comprehensively investigate the impact of various layer configurations, this section provides detailed results from a full set of ablation studies on layer selection. We explore multiple configurations, including individual layers as well as combinations of consecutive layers segmented into thirds and sixths of the model's depth, to verify the potential advantage of aggregating multiple consecutive layers for enhancing keyframe priors. This extended analysis further supports the observation discussed in Sec. 2.2, specifically highlighting that deeper attention layers yield more reliable keyframe priors.

| Method | Retention Ratio | CO | CU | CI | DO | TU | Average | |
|---|---|---|---|---|---|---|---|---|
| | | | | | | | Score | % |
| Vanilla | 100% | 3.11 | 3.17 | 2.79 | 2.83 | 2.20 | 2.82 | 100.0 |
| VisionZip | 50% | 2.91 | 3.08 | 2.72 | 2.74 | 2.03 | 2.70 | 95.7 |
| DyToK | | **3.04** | **3.15** | **2.79** | **2.80** | **2.17** | **2.79** | **98.9** |
| VisionZip | 25% | 2.62 | 2.86 | 2.50 | 2.42 | 1.81 | 2.44 | 86.5 |
| DyToK | | **3.01** | **3.11** | **2.76** | **2.68** | **2.10** | **2.73** | **96.8** |
| VisionZip | 15% | 2.20 | 2.48 | 2.14 | 2.08 | 1.61 | 2.10 | 74.5 |
| DyToK | | **2.91** | **3.02** | **2.67** | **2.54** | **1.97** | **2.62** | **92.9** |
| VisionZip | 10% | 1.46 | 1.85 | 1.51 | 1.58 | 1.24 | 1.53 | 54.3 |
| DyToK | | **2.62** | **2.75** | **2.38** | **2.24** | **1.84** | **2.37** | **84.0** |

Table 10: **Experiments on general descriptive queries.** We conduct experiments on VideoChatGPT using LLaVA-OneVision (7B) with 32 input frames, based on the VisionZip pruning method at retention ratios ranging from 10% to 50%.

| Method | Retention Ratio | Short | Medium | Long | Overall | Relative (%) |
|---|---|---|---|---|---|---|
| Vanilla | 100% | 69.9 | 56.7 | 48.9 | 58.5 | 100.0 |
| DyToK (w/ CLIP) | 25% | 68.3 | **56.9** | **49.1** | 58.1 | 99.4 |
| DyToK | | **69.2** | 56.8 | 48.9 | **58.3** | **99.7** |
| DyToK (w/ CLIP) | 15% | 64.6 | 53.9 | 48.4 | 55.6 | 95.1 |
| DyToK | | **66.3** | **55.0** | **48.7** | **56.7** | **97.0** |
| DyToK (w/ CLIP) | 10% | 57.6 | 50.9 | 45.8 | 51.4 | 87.9 |
| DyToK | | **61.0** | **51.1** | **47.6** | **53.2** | **91.0** |

Table 11: **Comparison of DyToK and CLIP-based keyframe selection.** We conduct experiments on LLaVA-OneVision (7B) using 32 input frames, with retention ratios ranging from 10% to 25%, based on the VisionZip pruning method.

Tab. 12 reports performance across alternative layer-selection strategies. We find that aggregating attention from the deepest third of layers (layers 16 to 23) achieves strong results, reaching 98.0% of baseline performance under a 20% retention ratio. Although its accuracy is slightly lower than that of the best-performing single layer (layer 20), its fixed percentage-based configuration generalizes better across models compared to manually tuning layer selection for each individual model. More broadly, accuracy improves with increasing layer depth: configurations leveraging later attention layers consistently exceed those restricted to earlier or shallower selections, indicating that temporally salient information is concentrated toward the top of the network. These findings substantiate the hypothesis that deeper attention layers more effectively capture relevant temporal cues under pruning-aware settings.

## A.7 Ablation on Token Allocation Upper Limit

As discussed in Appendix B.1 and Sec. 3.3, to ensure token budgets allocated using frame weights remain within reasonable bounds for each frame and mitigate the adverse effects of temporal attention outliers, we introduce a predefined upper limit for tokens per frame. If the allocated tokens exceed this upper limit, excess tokens are truncated and redistributed to other frames. This section comprehensively evaluates the influence of various upper limit settings on DyToK.

Tab. 13 presents detailed results across multiple upper limit configurations. Our findings reveal that setting a moderately restrictive upper limit significantly enhances performance. Particularly, an upper

| Layer | VideoMME | | | | LongVideoBench | MLVU | Average | |
|---|---|---|---|---|---|---|---|---|
| | Short | Medium | Long | Overall | | | Score | % |
| Vanilla (TMLR) | 70.0 | 56.7 | 48.8 | 58.5 | 56.6 | 47.1 | 54.1 | 100 |
| 0 | 62.6 | 53.1 | 46.4 | 54.0 | 52.0 | 40.8 | 48.9 | 90.5 |
| 4 | 63.6 | 53.3 | 49.0 | 55.3 | 53.5 | 42.3 | 50.4 | 93.1 |
| 8 | 64.1 | 53.4 | 47.3 | 55.0 | 53.4 | 41.5 | 50.0 | 92.3 |
| 12 | 65.7 | 54.9 | 47.8 | 56.1 | 53.3 | 43.3 | 50.9 | 94.1 |
| 16 | 66.2 | 55.3 | 48.2 | 56.6 | 53.4 | 44.9 | 51.6 | 95.4 |
| 20 | 68.1 | 56.6 | 47.8 | 57.5 | 55.5 | 46.3 | 53.1 | 98.2 |
| 23 | 66.3 | 56.3 | 48.7 | 57.1 | 53.8 | 45.1 | 52.0 | 96.1 |
| 0.7 | 64.7 | 53.3 | 48.4 | 55.5 | 53.5 | 43.7 | 50.9 | 94.0 |
| 8.15 | 66.0 | 55.4 | 48.0 | 56.5 | 54.8 | 44.5 | 51.9 | 96.0 |
| 16.23 | 69.2 | 56.4 | 47.8 | 57.8 | 56.0 | 45.2 | **53.0** | **98.0** |
| 0.3 | 63.2 | 54.0 | 48.7 | 55.3 | 54.2 | 42.4 | 50.6 | 93.5 |
| 4.7 | 65.6 | 53.8 | 48.8 | 56.0 | 54.5 | 43.9 | 51.5 | 95.2 |
| 8.11 | 66.9 | 55.3 | 49.7 | 57.3 | 55.2 | 44.9 | 52.5 | 97.0 |
| 12.15 | 66.3 | 57.1 | 48.3 | 57.3 | 53.8 | 41.6 | 50.9 | 94.1 |
| 16.19 | 67.0 | 56.2 | 46.7 | 56.6 | 54.8 | 42.7 | 51.4 | 94.9 |
| 20.23 | 69.3 | 55.7 | 48.7 | 57.9 | 54.5 | 46.1 | 52.8 | 97.6 |

Table 12: **Performance of DyToK under different layer configurations.** We conduct experiments using a retention ratio of 20%, with the number of frames set to 32. In comparison, the baseline LLaVA-OneVision retains its original configuration without any pruning or modification. The best result is highlighted in red, while the second-best is shown in **bold**.

| Upper Limit/Frame | VideoMME | | | | Egoschema | | LongVideoBench | MLVU | Average |
|---|---|---|---|---|---|---|---|---|---|
| | Short | Medium | Long | Overall | Subset | Total | | | |
| Vanilla (TMLR) | 70.0 | 56.7 | 48.8 | 58.5 | 62.0 | 60.0 | 56.6 | 47.1 | 56.6 |
| 196 | 69.2 | 56.4 | 47.8 | 57.8 | 62.2 | 59.8 | 56.0 | 45.2 | 54.7 |
| 176 | 69.0 | 56.3 | 47.9 | 57.7 | 62.2 | 59.8 | 55.8 | 45.0 | 54.6 |
| 147 | 68.7 | 56.3 | 48.1 | 57.7 | 62.4 | 59.8 | 56.2 | 44.7 | 54.6 |
| 118 | 68.0 | 56.0 | 48.0 | 57.3 | 62.0 | 59.9 | 55.9 | 44.7 | 54.5 |
| 98 | 68.4 | 55.8 | 47.9 | 57.4 | 62.8 | 59.9 | 55.8 | 46.6 | 54.9 |
| 78 | 68.2 | 55.3 | 49.0 | 57.5 | 62.0 | 59.8 | 56.4 | 45.5 | **54.8** |
| 49 | 68.0 | 56.7 | 49.0 | 57.9 | 63.2 | 60.2 | 55.1 | 45.2 | 54.6 |

Table 13: **Performance of DyToK under different upper limit configurations.** We conduct experiments using a retention ratio of 20%, with the number of frames set to 32. In comparison, the baseline LLaVA-OneVision retains its original configuration without any pruning or modification. The best result is highlighted in red, while the second-best is shown in **bold**.

| Method | Retention Ratio | VideoMME | LongVideo Bench | MLVU | Average | |
|---|---|---|---|---|---|---|
| | | | | | Score | % |
| Vanilla | 100% | 58.5 | 56.6 | 47.1 | 54.1 | 100.0 |
| Attention Entropy-Based | 25% | 58.3 | 55.9 | 45.6 | 53.3 | 98.5 |
| Feature Entropy-Based | | 58.3 | 54.8 | 42.9 | 52.0 | 96.1 |
| Feature Magnitude-Based | | 58.3 | **56.0** | 44.8 | 53.1 | 98.1 |
| DyToK | | **58.3** | 55.4 | **46.3** | **53.3** | **98.5** |
| Attention Entropy-Based | 15% | 55.2 | 52.8 | 43.4 | 50.5 | 93.2 |
| Feature Entropy-Based | | 55.2 | 52.7 | **44.9** | 50.9 | 94.1 |
| Feature Magnitude-Based | | 54.7 | 53.3 | 43.8 | 50.6 | 93.5 |
| DyToK | | **56.7** | **54.2** | 43.4 | **53.3** | **96.0** |
| Attention Entropy-Based | 10% | 50.6 | 48.1 | 38.6 | 45.7 | 84.5 |
| Feature Entropy-Based | | 50.7 | 48.8 | 38.4 | 45.9 | 84.9 |
| Feature Magnitude-Based | | 50.6 | 47.4 | 37.9 | 45.3 | 83.7 |
| DyToK | | **53.2** | **50.4** | **42.8** | **48.8** | **90.2** |

Table 14: **Ablation on importance estimation strategies.** We conduct experiments on LLaVA-OneVision (7B) using 32 input frames, with retention ratios ranging from 10% to 25%, based on the VisionZip pruning method.

limit of 98 tokens per frame achieves the highest average performance of 54.9, closely followed by the configuration with 78 tokens per frame at 54.8. These settings indicate that tighter upper limits efficiently suppress temporal attention outliers without excessively restricting token allocation to keyframes, thereby maintaining robust model accuracy.

## A.8   Ablation on Importance Estimation Strategies

To investigate the effects of various importance estimation strategies on the performance, we conduct ablation studies comparing our proposed cross-modal attention-based frame weighting method (utilized in the Temporal Importance Estimation stage) with entropy-based and magnitude-based approaches. Specifically, we explore the following alternative importance estimation strategies.

**Attention Entropy-Based:** For each frame, compute the attention weights of the last query token over all visual tokens. Calculate the entropy of these attention weights and normalize the results to obtain the frame weights.

**Feature Entropy-Based:** Compute the information entropy of visual tokens along the feature dimension. For each frame, average the entropy of all tokens and normalize the values to derive the frame weights.

**Feature Magnitude-Based:** Calculate the L2 norm of visual tokens for each frame. Average the norms and normalize the results to obtain the frame weights.

Tab. 14 shows that DyToK consistently outperforms entropy and magnitude-based methods across different retention ratios.

## A.9   Ablation on Textual Token Selection

To explore whether relying solely on the last textual token can fully capture the sequence's semantics, we conduct ablations on multiple long video benchmarks using LLaVA-OneVision (7B) under various retention ratios.

As shown in Tab. 15, at higher retention ratios (e.g., 75%, 50%), using only the last query token performs better. At more aggressive ratios (e.g., 25%, 15%), using all query tokens yields further gains. We hypothesize that the last token acts as a precise but narrow selector, while all tokens offer

| Method | Retention Ratio | VideoMME | LongVideo Bench | MLVU | Average Score | % |
|---|---|---|---|---|---|---|
| Vanilla | 100% | 58.5 | 56.6 | 47.1 | 54.1 | 100.0 |
| FastV | | 57.6 | 57.1 | 46.5 | 53.7 | 99.3 |
| DyToK (all query tokens) | 75% | 58.1 | **57.1** | 45.4 | 53.5 | 98.9 |
| DyToK (last query token) | | **58.4** | 56.8 | **46.8** | **54.0** | **99.8** |
| FastV | | 57.2 | 57.1 | 44.7 | 53.0 | 98.0 |
| DyToK (all query tokens) | 50% | 58.2 | **58.0** | 45.6 | 53.9 | 99.6 |
| DyToK (last query token) | | **58.5** | 57.2 | **46.3** | **54.0** | **99.8** |
| FastV | | 56.0 | 56.6 | 43.7 | 52.1 | 96.3 |
| DyToK (all query tokens) | 25% | **56.9** | **58.0** | 47.3 | **54.1** | **100.0** |
| DyToK (last query token) | | 56.3 | 55.7 | **47.8** | 53.3 | 98.5 |
| FastV | | 51.1 | 51.2 | 38.3 | 46.9 | 86.7 |
| DyToK (all query tokens) | 15% | 54.1 | **53.1** | **44.4** | **50.5** | **93.3** |
| DyToK (last query token) | | **54.8** | 52.6 | 43.2 | 50.2 | 92.8 |

Table 15: **Ablation on textual token selection.** We conduct experiments on LLaVA-OneVision (7B) using 32 input frames, based on the FastV pruning method.

broader coverage with higher recall. When visual evidence is abundant, precision reduces noise; when it is scarce, broader semantic coverage becomes essential.

# B  Detailed Analysis of Keyframe Prior in VLLMs

In this section, we provide an in-depth description and qualitative analysis of the keyframe prior phenomenon observed in VLLMs. We believe our findings will aid future research in understanding and addressing related challenges.

## B.1  Temporal Attention Outlier Phenomenon

**Motivation and methodology.**  As described in Sec. 2.2, effective token compression should dynamically allocate tokens according to each frame's relevance to the query, i.e., assigning more tokens to informative frames and fewer to less relevant ones. Accurate indicators of frame contribution are thus essential. We investigate whether the VLLM attention map can serve as such an indicator. Specifically, we compute attention weights from the last textual token to all visual tokens at each layer of the VLLM, producing fine-grained layer-frame correlations. Averaging these attention weights across frames and layers yields a temporal importance score, quantifying each frame's overall relevance. Similarly, averaging attention weights across the entire visual token sequence per layer indicates each layer's contribution to video comprehension.

**Experimental observations.**  We conduct experiments with 32-frame and 64-frame inputs, as illustrated in Fig. 5 and Fig. 6. Both LLaVA-OneVision [1] and LLaVA-Video [2] consistently exhibit accurate identification of keyframes relevant to specific queries. For example, when asked, "Which of the following outcomes occurs when the boy with brown hair and no glasses indicates the feather?", both models prominently attend to keyframe 23. However, we also notice a consistent bias toward disproportionately high attention allocation at the initial and final frames, regardless of their semantic irrelevance or even absence of meaningful information.

This phenomenon, termed *Temporal Attention Outlier*, becomes significantly pronounced with 64-frame inputs, wherein an additional, highly prominent outlier frame appears consistently in the middle of the sequence. Remarkably, this outlier frame remains constant across different queries for the same video and generally lacks relevant semantic content.

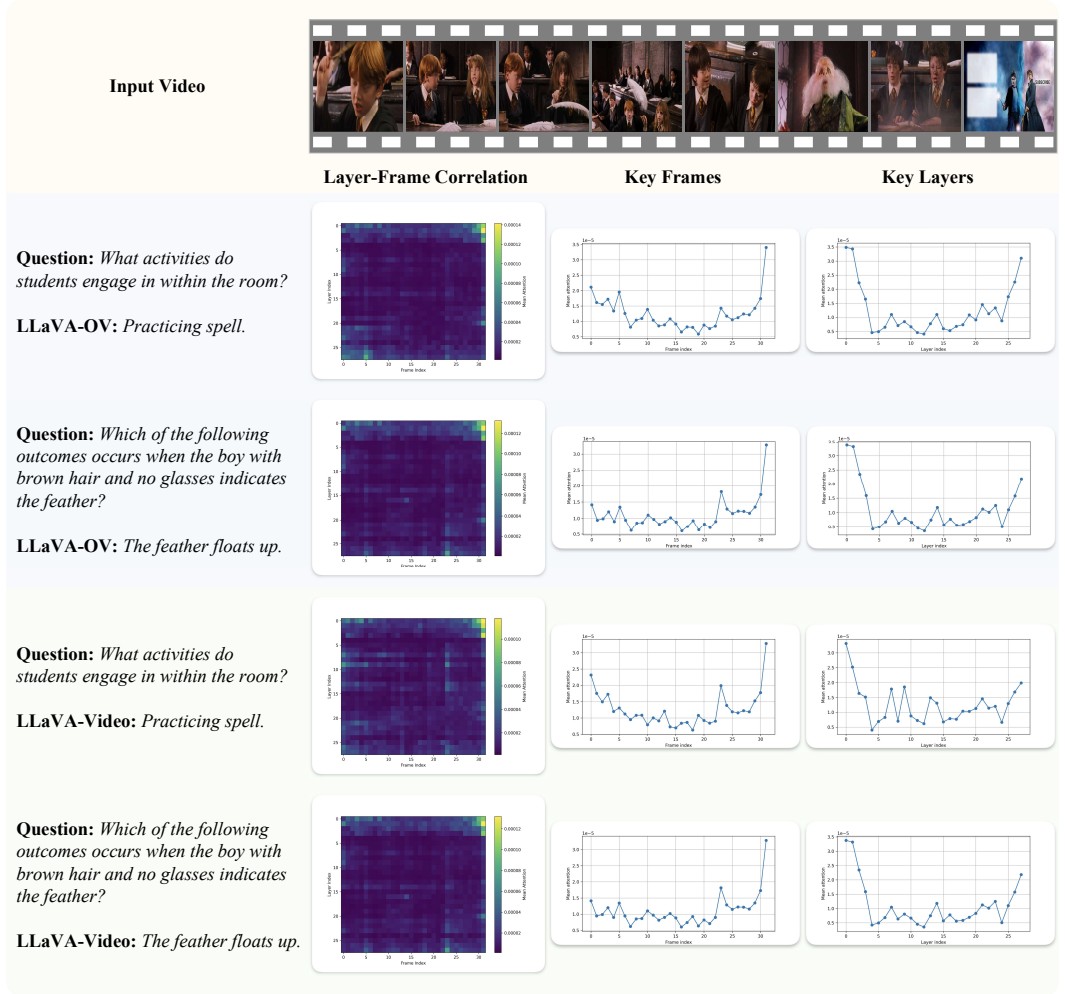

Figure 5: **Analysis of VLLMs with 32-frame inputs.** We visualize the attention behavior of LLaVA-OneVision and LLaVA-Video on a 32-frame video input under different queries. Each row shows the model's predicted answer, the layer-frame correlation heatmap, the frame-wise attention distribution, and the layer-wise attention weights. Both models exhibit consistent and accurate localization of task-relevant keyframes. However, we also observe a recurring bias toward the initial and final frames, where attention is disproportionately high despite their limited relevance.

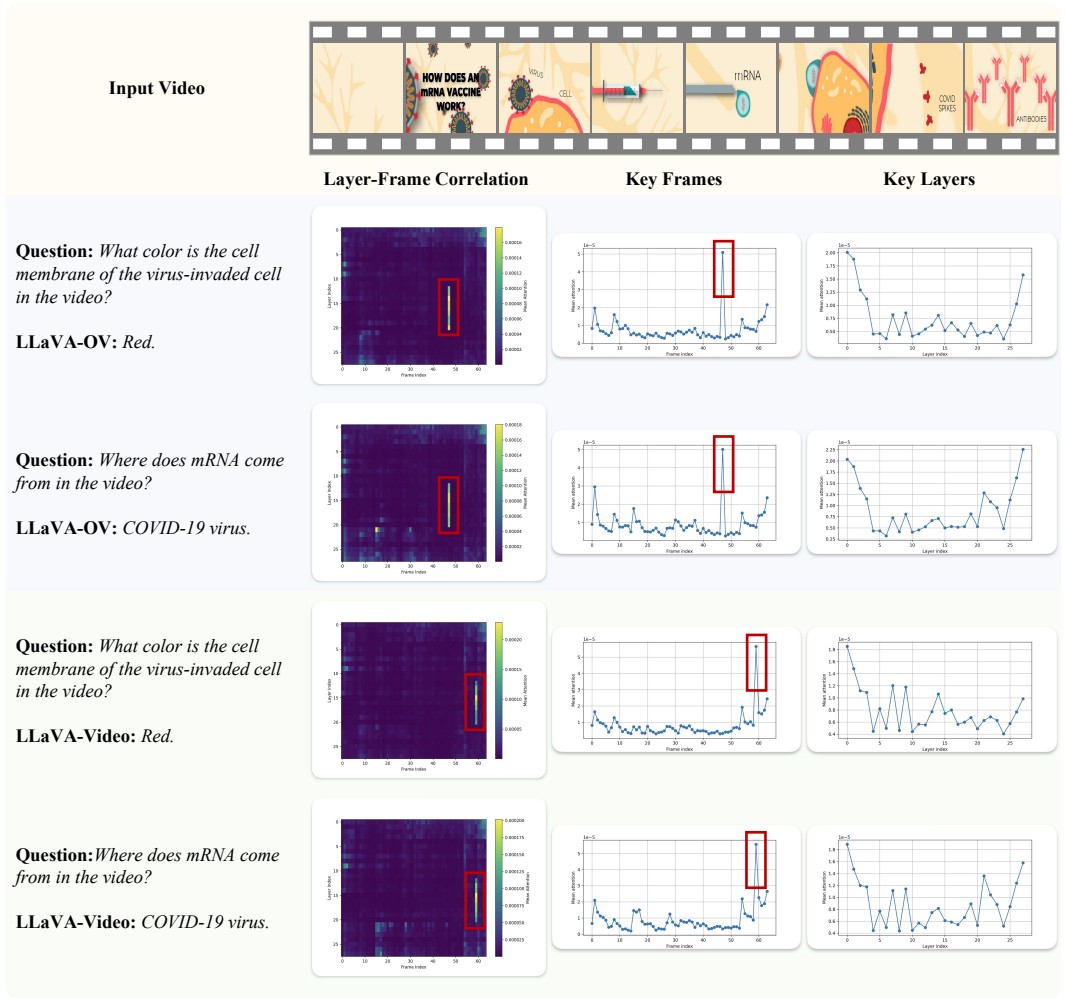

Figure 6: **Analysis of VLLMs with 64-frame inputs.** This figure presents the behavior of LLaVA-OneVision and LLaVA-Video on 64-frame inputs. Compared to the 32-frame case, the temporal attention bias becomes more prominent. In addition to edge-frame outliers, a strong and persistent attention peak emerges at a middle-frame location across different queries. This central outlier does not consistently contain meaningful visual content yet dominates attention allocation, revealing a new form of modality-invariant attention artifact we term as *Temporal Attention Outlier*.

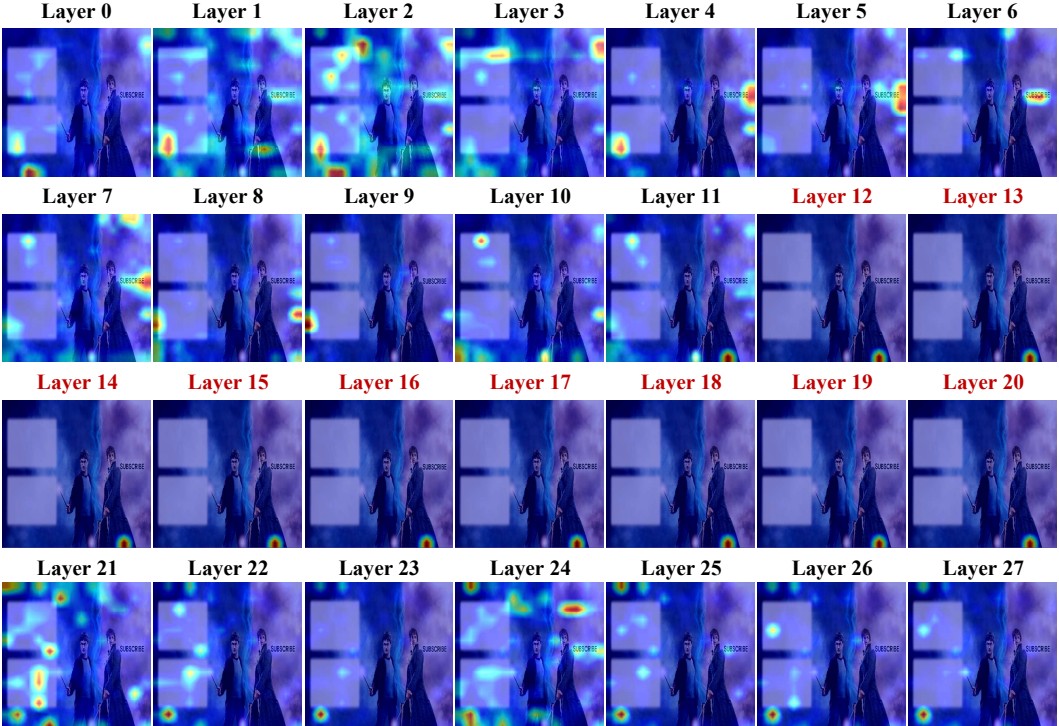

Figure 7: **Frame-level attention maps of LLaVA-OneVision across 28 layers.** Attention visualizations for the mid-sequence outlier frame reveal a pronounced and consistent activation pattern from layers 12 to 20, all focusing on the same non-semantic background patch. The similarity across layers reveals a copy-paste-like effect, indicating a temporal attention outlier that persists across the model hierarchy.

**Related work and interpretation.** Similar attention outlier phenomena have been observed in other domains. Registers [31] identify certain tokens in visual Transformers exhibiting extraordinarily high output norms despite lacking semantic content. Such tokens aggregate global information rather than local patch representations. Analogously, CLIPtrase [32] and DeCLIP [33] report "proxy/global token" effects in vision language models, where specific tokens gain disproportionate attention, diluting local semantic relationships and negatively impacting downstream dense prediction tasks. StreamingLLM [34] introduces the "attention sink" concept, observing structural anomalies whereby non-semantic initial tokens dominate attention distributions due to normalization constraints imposed by softmax operations.

Collectively, these studies highlight a general drawback in transformer attention mechanisms: certain non-informative tokens inevitably absorb disproportionate attention to maintain computational stability under softmax normalization. The temporal attention outlier phenomenon observed here is a temporal manifestation of this mechanism, distorting accurate attention allocation based on semantic relevance. This misallocation undermines the robustness and effectiveness of dynamic token compression strategies.

**Mitigation strategies.** Considering this detrimental effect, we suggest explicit strategies to mitigate attention outliers. As detailed in Sec. 3.3, imposing an upper limit $T_{max}$ on tokens per frame during budget allocation can effectively suppress outlier frames' influence by reallocating surplus tokens to genuinely important frames. Experimental results summarized in Tab. 13 demonstrate that lowering $T_{max}$ enhances post-pruning model accuracy.

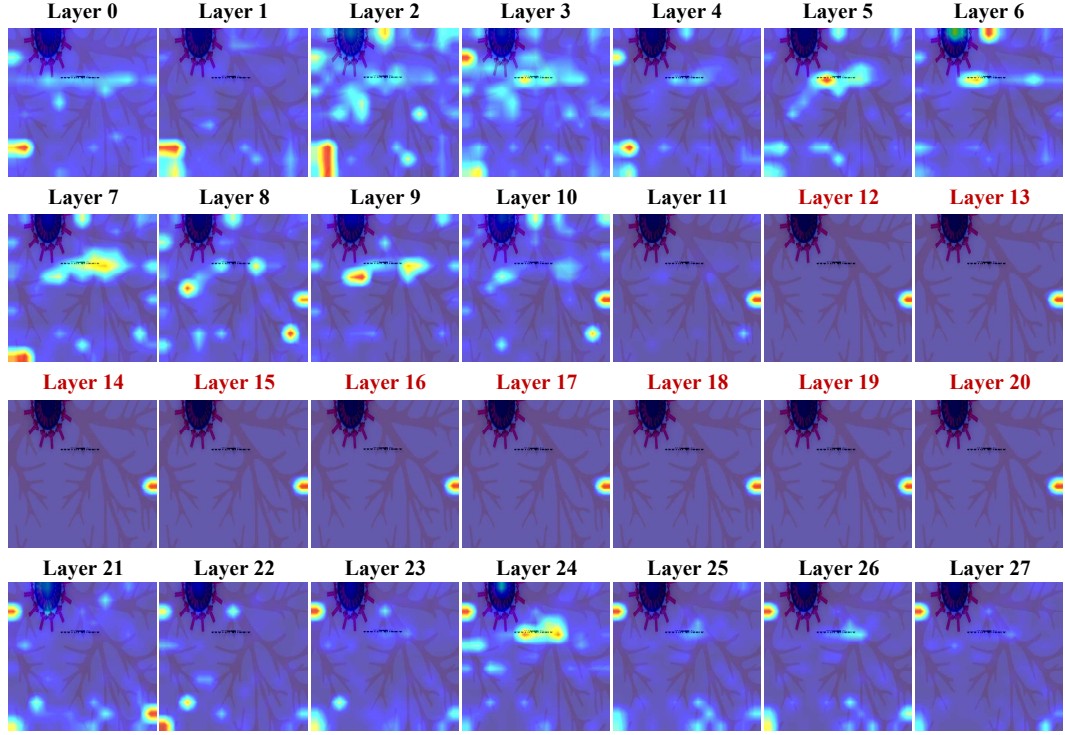

Figure 8: **Frame-level attention maps of LLaVA-Video across 28 layers.** Similar to LLaVA-OneVision, a distinct attention concentration is observed from layers 12 to 20, fixated on a spatially narrow and semantically irrelevant region. The uniformity of these patterns across layers highlights the presence of a systematic temporal attention outlier within VLLMs.

## B.2 Visualizing and Characterizing Temporal Attention Outliers

To further characterize temporal attention outliers, we visualize attention distributions at the frame-token level across layers (Fig. 7 and Fig. 8). For 64-frame inputs, a distinct bright horizontal line appears consistently at layers 12-20, corresponding precisely to the mid-sequence attention outlier frame. Detailed visualization reveals these layers exclusively attend to a single background patch within the outlier frame, with strikingly similar attention patterns across layers, resembling a "copy-paste" effect.

This comprehensive visualization reinforces the understanding that *temporal attention outliers consistently manifest at specific layers, target non-semantic regions, and represent a systematic rather than random anomaly within the VLLM architecture*.

## B.3 Exploring Keyframe Bias in VLLMs

The discovery of temporal attention outliers raises a fundamental question: Does an inherent bias toward specific frames ("keyframe bias") exist within VLLMs? To investigate this hypothesis, we present LLaVA-OneVision with semantically empty inputs, including black, white, and noise videos, matching the original video's frame count and resolution.

As illustrated in Fig. 9, significant attention peaks consistently appear at the initial, final, and mid-sequence frames, even without semantic content. This result strongly suggests an inherent structural bias within VLLMs toward these frames. Potential explanations include mechanisms similar to "attention sinks" [34], which manage excess attention distribution, or global patches [31, 32, 33, 35] used to aggregate information across sequences, facilitating processing under causal masking. Future research should further clarify these mechanisms.

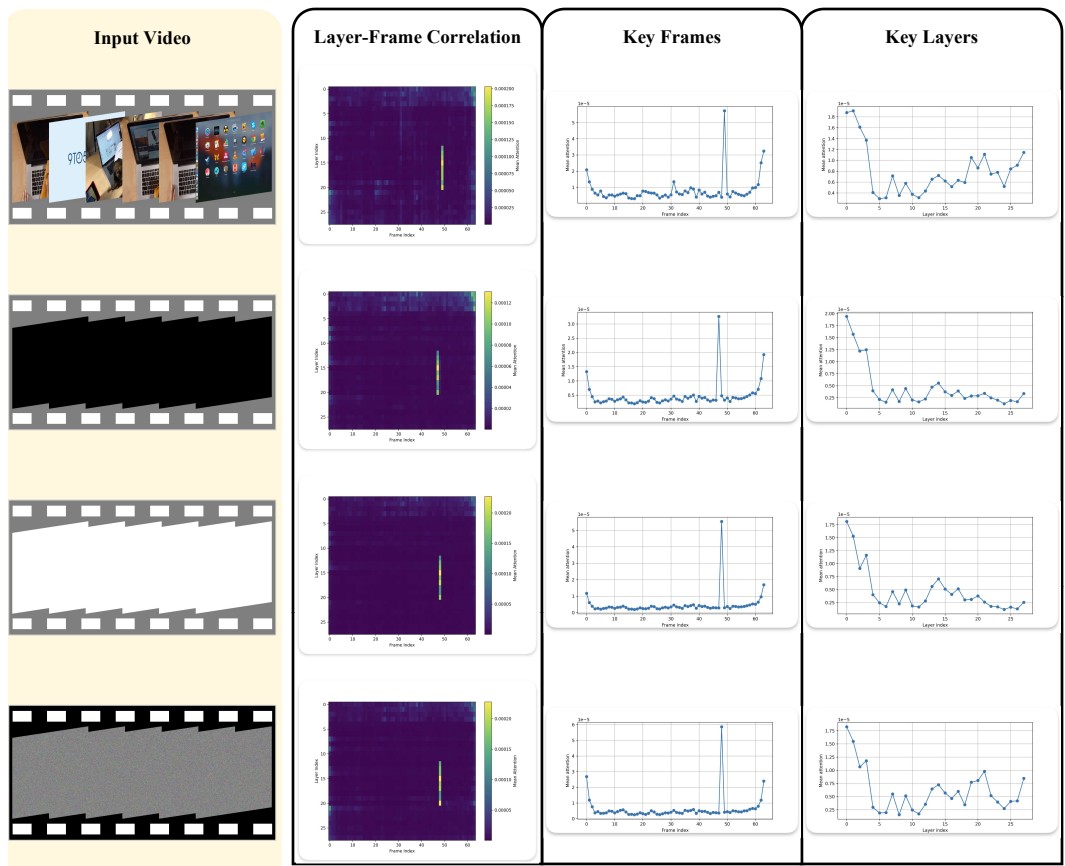

Figure 9: **Visualization of the keyframe bias in VLLMs.** Each row corresponds to a variant of synthetic input (black/white/noise videos) with minimal semantic content. The middle column presents the layer-frame correlation heatmaps, where a distinct vertical stripe indicates a mid-sequence attention outlier across layers 12 to 20. The right two plots show the corresponding frame-wise and layer-wise importance scores, with a sharp peak at the outlier frame. These consistent patterns across diverse non-semantic inputs reveal the systematic nature of keyframe bias in VLLMs.

We hope that our detailed exploration of keyframe bias and temporal attention outliers may provide useful insights for future research, contributing to more robust and interpretable VLLMs.

## C   Efficiency Analysis

In this section, we present additional experiments evaluating the efficiency improvements of DyToK, demonstrating significant reductions in GPU memory usage, FLOPs, and inference time with minimal impact on model accuracy.

**GPU memory usage.**   We first analyze the GPU memory efficiency of DyToK using LLaVA-OneVision (7B) on the VideoMME benchmark with 32-frame inputs. Experiments are conducted across various token retention ratios, ranging from 10% to 75%. As shown in Tab. 16, GPU memory consumption decreases substantially with lower retention ratios, achieving a maximum memory reduction of 77.9%. Memory savings are significantly influenced by model size, stabilizing around 17.3 GB at retention ratios of 50% and below, which approximates the minimal GPU memory footprint limited by the model's weights.

**Inference time analysis.**   To evaluate DyToK's inference efficiency, we measure the prefilling time on the VideoMME benchmark using LLaVA-OneVision (7B) with 32 input frames. As indicated in

| Method | Retention Ratio | FLOPs (T) | Memory (GB) | Latency (ms) | VideoMME | LongVideo Bench | MLVU | Avg. |
|---|---|---|---|---|---|---|---|---|
| Vanilla | 100% | 40.8 | 22.2 | 48.3 | 58.5 | 56.6 | 47.1 | 54.1 |
| DyToK | 75% | 32.6 | 20.2 | 44.1 | 59.0 | 55.9 | 46.6 | 53.8 |
| DyToK | 50% | 21.9 | 17.3 | 23.5 | 59.1 | 56.4 | 46.2 | 53.9 |
| DyToK | 25% | 12.2 | 17.3 | 11.3 | 58.3 | 55.4 | 46.3 | 53.3 |
| DyToK | 20% | 9.6 | 17.3 | 10.7 | 58.4 | 55.4 | 43.7 | 52.5 |
| DyToK | 15% | 8.4 | 17.3 | 10.4 | 56.7 | 54.2 | 43.4 | 51.4 |
| DyToK | 10% | 5.9 | 17.3 | 10.8 | 53.2 | 50.4 | 42.8 | 48.8 |

Table 16: **Analysis of DyToK on FLOPs, GPU memory, and prefilling time.** We conduct experiments on the LLaVA-OneVision model using 32 input frames, with pruning rates ranging from 10% to 75%, based on the VisionZip pruning method. For comparison, the baseline LLaVA-OneVision retains its original configuration without any pruning or modification.

Tab. 16, DyToK notably accelerates inference speed as the token retention ratio decreases. Specifically, at a 50% retention ratio, DyToK achieves a $2.1\times$ speedup, improving further to $4.3\times$ at a 25% retention ratio compared to the vanilla model.

**Accuracy-efficiency trade-off.** DyToK provides a balanced trade-off between accuracy and computational efficiency. At a moderate 50% retention ratio, DyToK retains 99.6% of the original model accuracy, accelerates inference by $2.1\times$, and reduces GPU memory usage by 22.1%. At a lower retention ratio of 25%, DyToK maintains 98.5% accuracy alongside a $4.3$ inference speedup. Even at the extreme retention ratio of 10%, equivalent to retaining approximately 14 tokens per frame, DyToK still preserves 90.2% accuracy, underscoring its robustness under aggressive compression.

# D  Related Works

**Vision-Language Models.** Inspired by the success of large language models (LLMs) [36, 37, 38, 39], recent Vision–Language Models (VLMs) combine a visual encoder [16, 17, 40, 41] with a pretrained LLM [37, 36] to enable large-scale multimodal reasoning. Pioneering systems such as BLIP-2 [42] and LLaVA [43] pass visual features through a lightweight projector before concatenating with textual tokens. While effective, this design suffers from a significant token-length bottleneck: a single high-resolution image ($672\times672$ in LLaVA-NeXT [44]) yields over 2,000 patch tokens, overwhelming the textual prompt. Multi-image or video inputs [1, 2, 3] further increase the sequence length. Therefore, maintaining model performance under a constrained visual token budget is critical for the continued advancement of VLMs.

**Efficient Vision-Language Models.** In the development of VLMs, two main approaches have been proposed to accelerate inference, targeting the two core components of the system. One common strategy involves pruning on the encoder side [12, 45, 46, 47, 48]. These methods typically identify dominant tokens and merge contextual tokens to preserve information from pruned tokens. Another prevalent approach focuses on pruning within the LLM itself. For example, FastV [8] selects important vision tokens based on attention scores but neglects the contribution of low-attention tokens. SparseVLM [22] incorporates text guidance via cross-modal attention, yet faces similar limitations. DART [49] addresses this by considering not only high-priority tokens but also applying token reduction techniques to retain information from low-attention tokens. These methods collectively aim to reduce the KV cache size and improve inference efficiency in LLMs.

**Efficient Video-Language Models.** Due to the large number of vision tokens in video inputs, recent video compression methods [10, 13, 50] have attracted growing attention. In the early stages, most pruning techniques were adapted from image-based approaches, which inevitably overlooked the temporal dependencies inherent in video data. For example, PruneVID [10] clusters video tokens and selects those most relevant to the query tokens, yet it fails to fully capture temporal continuity. More

recent approaches have begun to address this limitation. FastVID [50], for instance, incorporates both temporal and visual context to perform spatiotemporal token pruning at the input stage, offering a more holistic approach to video token reduction.

**Frame Selection Methods.** Given the varying information density across video frames, most existing VLLMs [3, 1, 2, 20] adopt fixed-duration sampling strategies. However, this uniform sampling often leads to sub-optimal performance, as it fails to account for the relevance of specific frames to the query. Several recent works [51, 52, 53] aim to improve frame selection by clustering input frames to minimize redundancy, but these methods typically incur high computational overhead during the selection process. An alternative line of work [54] dynamically allocates tokens to each frame based on their query relevance. QuoTA [54], for instance, employs a lightweight model to evaluate the relationship between each frame and the query in order to assign token budgets, but this introduces significant latency prior to inference. In contrast, our approach eliminates redundancy before inference by leveraging attention weight scores as indicators for token allocation.

# E  Efficient inference paradigms for VLLMs

Recent advancements in efficient inference for VLLMs can be categorized into two paradigms: LLM attention-based token pruning [10, 13] and encoder feature-based token selection [23, 24]. These paradigms differ fundamentally in their operational stages, criteria for token importance, and stability under varying conditions.

*LLM Attention-Based Token Pruning.* As shown in Fig. 2(a), this paradigm dynamically prunes redundant visual tokens during the LLM inference stage. It operates by ranking token importance at intermediate transformer layers within the LLM, typically using attention scores as criteria. Tokens with lower importance scores are progressively discarded in subsequent layers. Theoretically, the computational cost reduction can be modeled as:

$$1 - \frac{\sum_{l=1}^{K} C(n_l) + \sum_{l=K+1}^{T} C(\hat{n}_l)}{\sum_{l=1}^{T} C(n_l)}, \qquad (6)$$

where $C(n)$ denotes the FLOPs of a transformer layer processing $n$ tokens, $K$ is the pruning start layer, and $\hat{n}_l \leq n_l$ represents the retained token count after pruning.

While this approach aligns token retention with task-specific textual prompts, its efficacy hinges critically on the quality of attention maps from designated pruning layers. Shallow layers often produce noisy attention signals due to insufficient semantic integration, whereas deeper layers yield delayed pruning decisions that diminish computational savings. This layer-specific dependency introduces instability, particularly for long video sequences requiring multi-layer reasoning.

*Encoder Feature-Based Token Selection.* Differently, this paradigm reduces redundancy at the visual encoder's output stage by selecting tokens based on intrinsic feature importance before feeding them to the LLM, as depicted in Fig. 2(b). Importance is quantified through static criteria derived from encoder self-attention patterns or feature activations, such as the average attention received by each token across all spatial or temporal positions. The token set is then truncated to a subset of size $k \ll n$, achieving a theoretical FLOPs reduction ratio of:

$$1 - \frac{C_{\text{proj}+\text{LLM}}(k)}{C_{\text{proj}+\text{LLM}}(n)}, \qquad (7)$$

where $C_{\text{proj}+\text{LLM}}(\cdot)$ encompasses both projector and LLM computations. This paradigm avoids layer-specific dependencies by leveraging stable encoder-derived features, ensuring consistent token selection regardless of downstream LLM depth.

Our method can be seamlessly integrated into both of these VLLM inference acceleration paradigms in a training-free and plug-and-play manner. By leveraging the inherent keyframe prior within VLLMs, DyToK provides temporal frame-importance weights to guide underlying token compression strategies. This allows more tokens to be retained for keyframes that are critical to the current task while reducing token allocation for redundant frames. Under limited token budgets, DyToK helps focus the model's attention on truly informative keyframes and enhances the temporal perception capabilities of VLLMs.

# F    Evaluation Benchmarks

We conducted experiments on these widely used long video understanding benchmarks.

**VideoMME.**    VideoMME [6] comprises 2,700 human-annotated multiple-choice questions drawn from 900 videos ($\approx$ 254h) across 6 visual domains and 30 sub-fields; the evaluation therefore covers a wide spectrum of spatial scenes and temporal durations, probing models' ability to track multimodal cues (frames + audio + subtitles) from 11-second clips up to 1-hour stories.

**LongVideoBench.**    LongVideoBench [7] comprises 6,678 human-curated multiple-choice questions built on 3,763 web videos (up to 1h) and their subtitles, organized into 17 fine-grained categories that revolve around a novel "referring-reasoning" task, thereby requiring models to localize, retrieve, and integrate visual-linguistic evidence over very long spatial–temporal contexts.

**EgoSchema.**    EgoSchema [55] comprises around 5,000 five-option multiple-choice questions collected from 250h of egocentric footage, emphasizing very long-form temporal reasoning—each three-minute clip demands tracing objects and actions over intrinsic time spans that are 5-10× longer than prior datasets, thereby stressing both spatial perception and extended temporal coherence.

**MLVU.**    MLVU [27] comprises 3,102 multiple-choice questions that span 9 distinct long-video understanding tasks, challenging models to handle videos ranging from 3 minutes to 2 hours and to reason over plot, temporal order, event retrieval, and other facets that jointly test fine-grained spatial recognition and long-range temporal logic.

# G    Implementation Details

## G.1    Experimental Settings

The vanilla LLaVA-OneVision employs SO400M [17] as its vision encoder and Qwen2 [37] as its language model. The bilinear pooling strategy reduces the number of vision tokens per frame from 729 to 196. We use two settings for the total number of frames—32 and 64—consistent with prior work [56, 57]. This results in 6,272 tokens for 32 frames and 12,544 tokens for 64 frames.

To validate the generalization ability of DyToK on different VLLM inference acceleration paradigms, we integrate DyToK into the existing SOTA methods, FastV [8], VisionZip [12], and DyCoke [13], which perform visual token pruning inside LLM, between the visual encoder and LLM, and on both sides, respectively. Specifically, FastV performs token pruning based on the text-to-visual attention extracted from a specific layer in LLM, VisionZip utilizes inter-patch feature correlations to apply token pruning after the visual encoder and before LLM, while DyCoke prunes visual tokens before and after entering into LLM, where the Token Temporal Merging (TTM) strategy compresses the redundancy in the temporal dimension by token merging on the encoder side, and the Dynamic Pruning (DP) strategy prunes visual tokens in the decoding stage based on the last query-to-visual attention on the LLM side.

## G.2    Token Budget Alignment

To ensure a fair comparison, we limit all methods to the same token budget. Based on this budget, we calculate the actual compression ratios specific to each one of FastV, DyCoKe, and VisionZip, so that the FLOPs of the computation of visual tokens during inference remain strictly consistent across methods.

To this end, we use a simple yet effective strategy to align the computational cost of different pruning strategies by keeping the total number of visual tokens processed by LLM across all layers the same. Assuming an LLM with $L$ layers, token pruning occurs at layer $K$. Let $N$ represent the initial number of visual tokens per layer before pruning, and $M$ denote the number of tokens retained after pruning at layer $K$. To quantify the average computational cost per layer, we introduce $R$, satisfying the following equation:

$$K \times N + (L - K) \times M = L \times R. \tag{8}$$

Thus, two methods can be considered computationally equivalent if their respective $R$ values match.

| Dominant | Contextual | Retention Ratio | Numerical Sum | Approximate Sum |
|---|---|---|---|---|
| 168 | 28 | 100% | 196 | 196 |
| 126 | 21 | 75% | 147 | 147 |
| 84 | 14 | 50% | 98 | 98 |
| 42 | 7 | 25% | 49 | 49 |
| 30 | 5 | 20% | 39.2 | 35 |
| 24 | 4 | 15% | 29.4 | 28 |
| 12 | 2 | 10% | 19.6 | 14 |

Table 17: **Token budget allocation under different retention ratios in DyToK-enhanced VisionZip.** A fixed 6:1 ratio is maintained between dominant and contextual tokens. The total number of retained tokens is rounded down to the nearest multiple of 7 to meet this ratio, resulting in lower actual token counts, especially under extreme compression.

**Adaptation of VisionZip.**   When integrating DyToK into VisionZip, to standardize hyperparameter settings across experiments, we maintain a fixed dominant-to-contextual token ratio of 6:1 per frame. Given that LLaVA-OneVision originally processes 196 tokens per frame, exact division isn't feasible, necessitating approximate rounding. To deliberately introduce a challenging scenario highlighting DyToK's robustness, we uniformly round down token counts to multiples of seven (see Tab. G.2). Token budgets used in subsequent methods are also aligned with the values presented in this table. The rounding strategy significantly reduces the effective token budget, particularly at extreme compression levels. For instance, at a 10% retention ratio, this rounding strategy yields approximately 40% fewer tokens compared to standard calculations without rounding. Consequently, accuracy scores under identical nominal retention ratios in this paper may appear slightly lower than those reported in comparable literature. However, this deliberate choice does not obscure the clear superiority of DyToK's performance.

**Adaptation of DyCoke.**   We hypothesize that token pruning in LLM only occurs at the prefilling stage for computational overhead alignment. However, DyCoke doesn't perform token pruning in LLM during the prefilling phase in order to obtain the KV Cache of all visual tokens across all layers for dynamic pruning in the decoding stage. The prefilling stage is crucial for some benchmarks like VideoMME in which LLM generates short responses (*e.g.* one token). Thus, we transform the dynamic pruning strategy in the second stage of DyCoke into static and only perform pruning in the prefilling stage for fair comparison.

**Adaptation of token compression methods to Qwen2.5-VL.**   Qwen2.5-VL adopts a sliding-window computation strategy that supports video frames of arbitrary resolution. At the end of its encoder, it introduces a patch merger module that reduces computational overhead by merging every four spatially adjacent visual tokens into one. However, this architecture poses significant challenges for the adaptation of token compression methods designed for other VLLMs.

Unlike the common 2D pooling techniques (e.g., bilinear or average pooling) used in models like LLaVA-OneVision and LLaVA-Video [2], Qwen2.5-VL [3] performs token merging by reshaping the token layout and feeding it into an MLP to fuse features in the embedding space. To accommodate this mechanism in encoder feature-based methods such as VisionZip, we retain the MLP-based merging strategy for the hidden states. However, to align the attention weights and importance scores, originally computed before merging with the post-merging hidden states, we apply spatially localized averaging to these metrics using the same compression ratio.

In addition, Qwen2.5-VL employs Multimodal Rotary Position Embedding (MRoPE), a positional encoding scheme that encodes each visual token based on its precise temporal, height, and width indices. Compression methods like VisionZip, which discard original token positions during pruning, are therefore inherently incompatible with mRoPE. To address this issue, we degrade the multimodal mRoPE to a standard 1D RoPE, consistent with the approach used for LLaVA-OneVision, ensuring compatibility without altering the model architecture.

# H    Visualization of Per-Frame Token Compression

To intuitively illustrate the effectiveness of dynamic token compression via LLM-guided keyframe prior, we visualize per-frame compression results on video-question pairs randomly sampled from the widely used VideoMME [6] benchmark. Each input video is uniformly sampled to 8 frames, with the ground-truth keyframes marked by red pentagrams and all other frames denoted by yellow circles in temporal order.

Fig. 10, Fig. 11, Fig. 12, and Fig. 13 present results for an encoder feature-based method, VisionZip [12], under an extreme token retention ratio of 10%. When the token budget is uniformly allocated across frames, VisionZip fails to sufficiently preserve information from critical keyframes while retaining excessive tokens from less relevant frames. This leads to substantial semantic distraction and ultimately incorrect answers. In contrast, our DyToK-enhanced version leverages LLM-guided keyframe priors to assign token budgets based on frame-level importance. As a result, DyToK effectively prioritizes tokens for keyframes that are most relevant to the question, while aggressively compressing redundant frames, leading to enhanced temporal perception under constrained computational budgets.

Fig. 14, Fig. 15, Fig. 16, and Fig. 17 show similar comparisons for an LLM attention-based method, FastV [8], evaluated under a 15% retention ratio. Uniform token allocation again results in a loss of key semantic content and an accumulation of redundant context. DyToK consistently mitigates these issues by dynamically adjusting token budgets across frames, demonstrating its general applicability across different VLLM inference acceleration paradigms.

To accurately illustrate the actual attention distribution guided by the keyframe prior in the token budget allocation process, we applied smoothing techniques to certain temporal attention outliers, details of which can be found in Appendix B.1. Importantly, DyToK is designed as a training-free, lightweight, and pluggable module capable of dynamically assigning compression ratios based on frame-query relevance. Our visualizations clearly demonstrate its effectiveness in dynamically allocating token budgets. However, the precise distribution and exact positions of retained tokens within individual frames depend on the underlying pruning method, such as VisionZip or FastV.

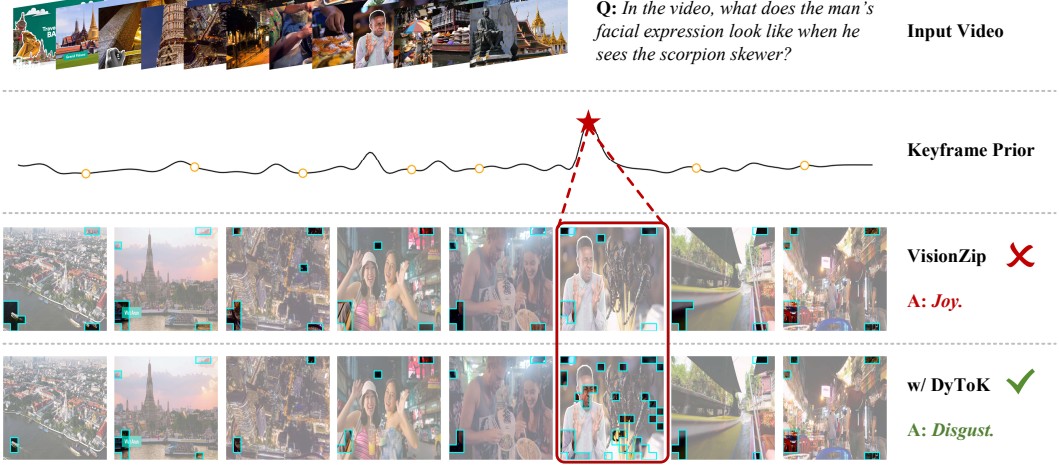

Figure 10: **Example 1 comparing VisionZip and VisionZip w/ DyToK under 10% retention ratio.** The DyToK-enhanced method better preserves keyframe tokens and suppresses temporal redundancy under the limited budget.

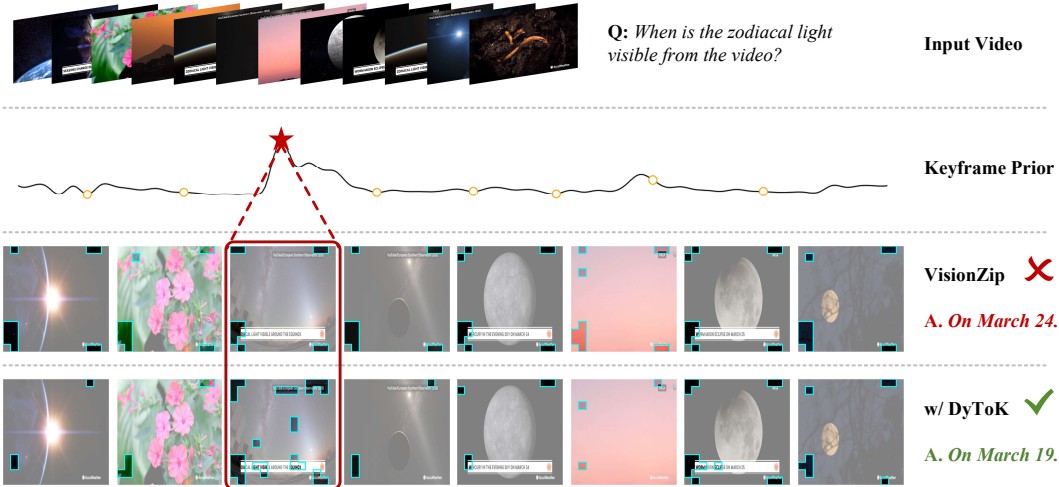

Figure 11: Example 2 comparing VisionZip and VisionZip w/ DyToK under 10% retention ratio.

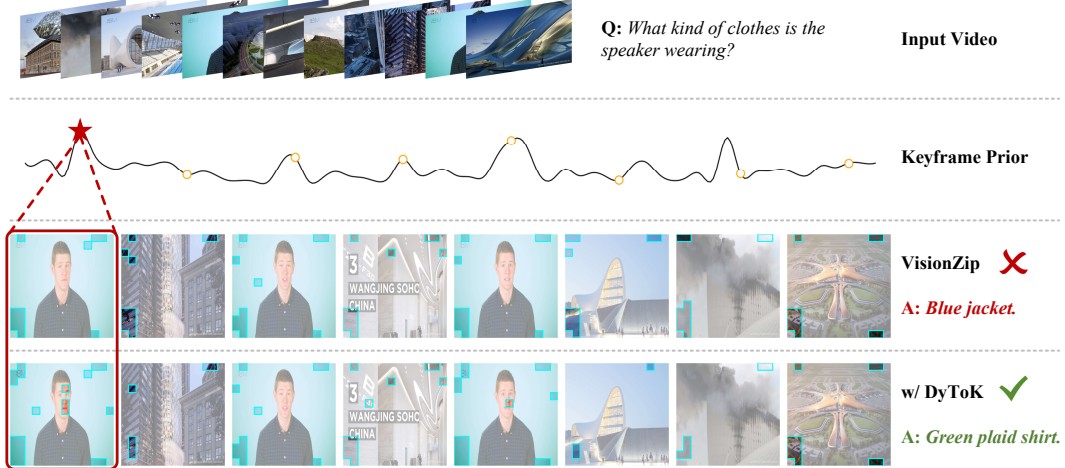

Figure 12: Example 3 comparing VisionZip and VisionZip w/ DyToK under 10% retention ratio.

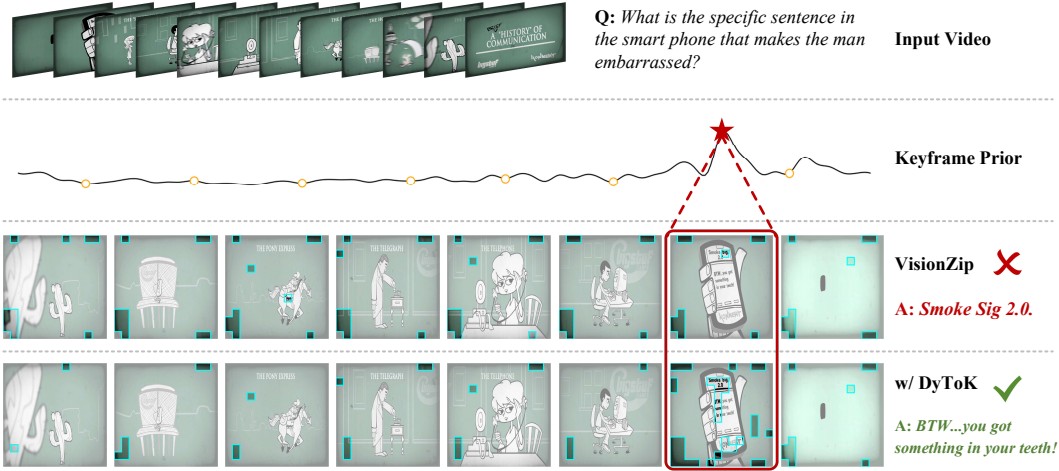

Figure 13: Example 4 comparing VisionZip and VisionZip w/ DyToK under 10% retention ratio.

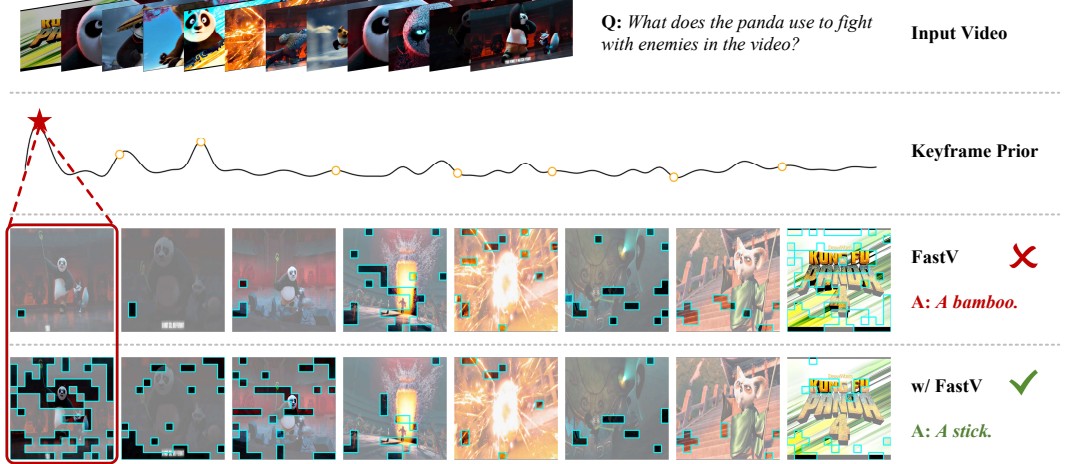

Figure 14: **Example 1 comparing FastV and FastV w/ DyToK under 15% retention ratio.** The DyToK-enhanced variant achieves better focus on keyframe regions and reduces attention to temporally redundant content under the constrained token budget.

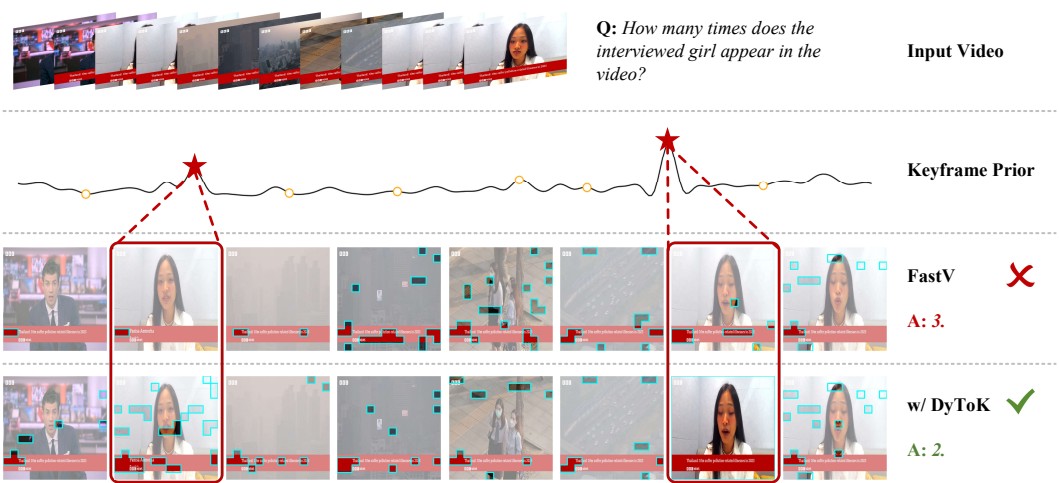

Figure 15: Example 2 comparing FastV and FastV w/ DyToK under 15% retention ratio.

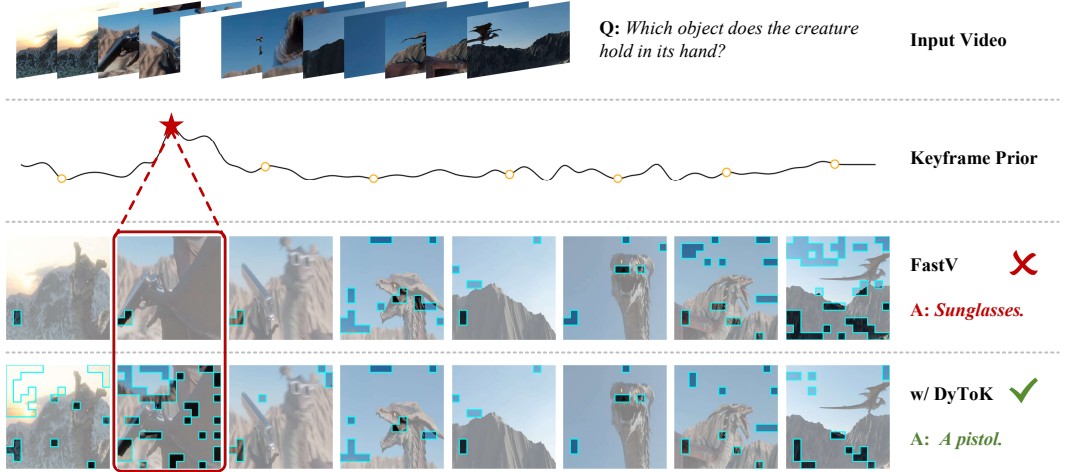

Figure 16: Example 3 comparing FastV and FastV w/ DyToK under 15% retention ratio.

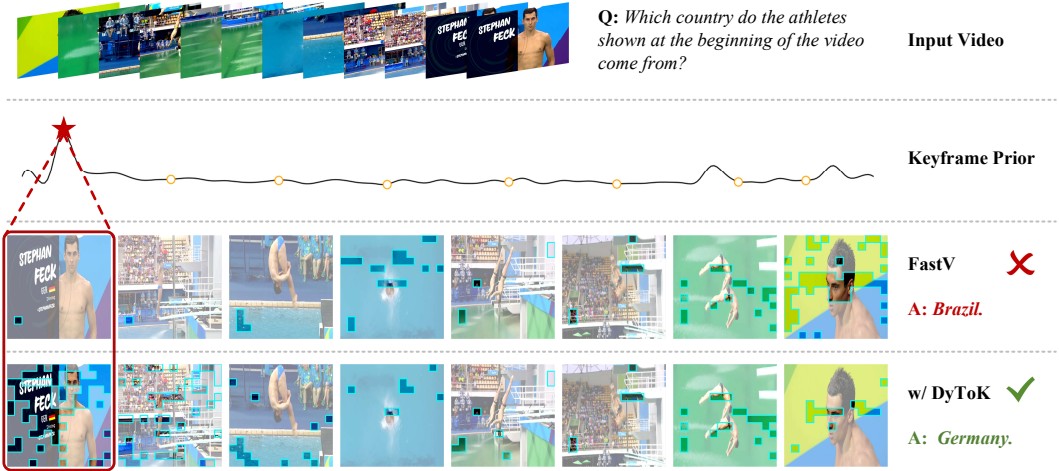

Figure 17: Example 4 comparing FastV and FastV w/ DyToK under 15% retention ratio.

