# OpenReview forum: "Less Is More, but Where? Dynamic Token Compression via LLM-Guided Keyframe Prior"
_NeurIPS.cc/2025/Conference — NeurIPS 2025 poster_

### Official Review · Reviewer_bZfF · 2025-06-26

**Clarity:** 2
**Significance:** 3
**Originality:** 3
**Rating:** 4
**Confidence:** 4

**Summary:**

This paper addresses the inefficiency of Video Large Language Models (VLLMs) in processing long videos, where computational costs rise due to redundant visual information. The authors propose DyToK, a training-free paradigm for dynamic token compression during inference. It leverages attention scores from VLLMs to distinguish between key frames and less relevant ones, adapting each frame’s token retention ratio based on its estimated importance. The method is plug-and-play compatible with existing compression techniques and applicable across multiple VLLM families. The authors conduct extensive experiments, visualizations, and analyses, showing that DyToK effectively identifies relevant key frames and achieves superior efficiency–accuracy trade-offs across various video benchmarks.

**Questions:**

1. The auxiliary lightweight assistant model used to obtain attention maps is derived from the same architectural family as the main model. However, I believe its feasibility and generalizability require further clarification. a) In addition to the final benchmark results, it would be helpful to include visualizations or statistical analyses. b) This design choice may also limit the method’s generalizability, especially for models that lack a corresponding lightweight version or where the lightweight version differs significantly in architecture. c) What would happen if a lightweight model from a different architectural family were used?
2. The practical benefits of reducing the number of tokens can be better demonstrated. For example, the actual speed-up in inference time with the same number of input frames, or the performance gains under a fixed computational budget.
3. Although the authors conducted extensive experiments and analyses, the presentation could be improved. For instance, performance trends across different token compression ratios could be presented more clearly, and visualizing the keyframe scores more intuitively in the main text would help better support the proposed method.
4. a) Using only the last token of the query to score the frames may overlook some important information. b) If the query does not explicitly point to specific visual content (e.g., a general descriptive query), can this method still effectively select important frames in the video?

My overall score may be adjusted based on the authors’ responses to these questions.

**Ethical Concerns:**

["NO or VERY MINOR ethics concerns only"]

**Final Justification:**

My initial judgment was borderline. The authors have provided additional results that reveal the underlying relationships between attention patterns across different architectures and models, which support the proposed method. Although my assessment remains borderline, I now lean toward the positive side, and my final rating is BA.

**Limitations:**

In the limitations section, the authors mention that, to balance performance and efficiency, they introduced an auxiliary lightweight assistant model to approximate the original model. I believe the effectiveness and generalizability of this approach require further analysis and discussion, as noted in the questions above.

**Quality:**

2

**Strengths And Weaknesses:**

Strengths:
- Quality: The proposed method is technically sound and thoroughly evaluated through extensive experiments across multiple VLLMs and token compression techniques. The authors also provide visualizations and analyses to support their claims.
- Clarity: The paper is clearly written and easy to follow. Technical details are well-explained, with helpful insights that facilitate a clear understanding.
- Significance: The work addresses the critical challenge of improving efficiency in long-video understanding for VLLMs. The analysis of attention maps across different layers for keyframe identification is insightful and potentially valuable for future research.
- Originality: The training-free, plug-and-play paradigm and its seamless integration with existing compression techniques offer a novel and practical perspective on advancing token compression in VLLMs.

Weakness:
- Quality: The method relies on an auxiliary lightweight model from the same architectural family to approximate the original model. The effectiveness and generalizability of this approach are not fully validated and warrant further analysis and discussion.
- Clarity: There are several noticeable typos in the paper, such as the placeholder “on XX benchmarks” in Line 65. The authors should proofread the manuscript more carefully.
- Significance: The practical benefits of token reduction, such as inference speed-up or extended video support under fixed computational budgets, are not sufficiently demonstrated in the paper.

---

> ### Author Rebuttal · Authors · 2025-07-31
>
> We sincerely appreciate your insightful comments. All suggestions will be incorporated into the paper, and typos will be fixed. Our responses are as follows.
>
> &nbsp;
> > **Response to Weakness 1 & Question 1(b, c) (Concern about Cross-Architecture Transferability)**
>
> To better demonstrate DyToK's transferability, we conduct experiments on VideoMME, using assistant and primary models with distinctly different architectures under varying retention ratios.
>
> To maximize the differences in architectures and thus rigorously test DyToK's robustness, we select LLaVA-OneVision (OV) and Qwen2.5-VL (Qwen), two open-source frameworks exhibiting significant architectural differences.
>
> Preliminary results are shown below, with OV using 0.5B/7B and Qwen using 3B/7B as assistant and primary models.
>
> |Method|Retention Ratio|Short|Medium|Long|Overall|Relative Acc. (%)|
> |---|---|---|---|---|---|---|
> |Vanilla (OV)|N/A|69.9|56.7|48.9|58.5|100.0|
> |VisionZip|50%|66.8|56.6|48.2|57.2|97.8|
> |DyToK (OV guides OV)|50%|**71.9**|56.1|**49.3**|**59.1**|**101.0**|
> |DyToK (Qwen guides OV)|50%|71.6|**56.6**|48.1|58.7|100.3|
> |VisionZip|25%|62.2|53.0|47.4|54.2|92.6|
> |DyToK (OV guides OV)|25%|69.2|**56.8**|**48.9**|58.3|99.7|
> |DyToK (Qwen guides OV)|25%|**71.3**|56.6|48.1|**58.9**|**100.7**|
> |VisionZip|20%|60.2|51.0|45.7|52.3|89.4|
> |DyToK (OV guides OV)|20%|**69.2**|**56.4**|47.8|**57.8**|**98.8**|
> |DyToK (Qwen guides OV)|20%|68.7|55.9|**48.2**|57.6|98.5|
> |VisionZip|15%|55.9|49.9|44.3|50.0|85.5|
> |DyToK (OV guides OV)|15%|66.3|**55.0**|48.7|**56.7**|**96.9**|
> |DyToK (Qwen guides OV)|15%|**67.3**|53.7|**48.9**|56.6|96.8|
> |VisionZip|10%|47.0|44.2|42.2|44.5|76.1|
> |DyToK (OV guides OV)|10%|**61.0**|51.1|47.6|53.2|90.9|
> |DyToK (Qwen guides OV)|10%|59.2|**53.3**|**47.9**|**53.5**|**91.5**|
>
> |Method|Retention Ratio|Short|Medium|Long|Overall|Relative Acc. (%)|
> |---|---|---|---|---|---|---|
> |Vanilla (Qwen)|N/A|72.6|61.4|49.9|61.3|100.0|
> |FastV|50%|63.9|52.7|48.0|54.9|89.5|
> |DyToK (Qwen guides Qwen)|50%|**72.0**|60.2|**51.7**|**61.3**|**100.0**|
> |DyToK (OV guides Qwen)|50%|71.8|**60.4**|51.0|61.1|99.6|
> |FastV|25%|59.9|49.1|46.8|51.9|84.7|
> |DyToK (Qwen guides Qwen)|25%|**71.7**|**58.9**|51.3|**60.6**|**98.9**|
> |DyToK (OV guides Qwen)|25%|71.3|58.3|**51.8**|60.5|98.7|
> |FastV|20%|57.8|48.4|45.6|50.6|82.5|
> |DyToK (Qwen guides Qwen)|20%|**70.8**|**57.7**|**51.4**|**60.0**|**97.8**|
> |DyToK (OV guides Qwen)|20%|**70.8**|57.2|51.2|59.7|97.5|
> |FastV|15%|54.9|45.0|44.2|48.0|78.4|
> |DyToK (Qwen guides Qwen)|15%|**68.4**|**56.3**|**50.1**|**58.3**|**95.1**|
> |DyToK (OV guides Qwen)|15%|67.2|55.1|49.7|57.3|93.5|
>
> Results show that DyToK remains robust even with cross-architecture assistant-primary model pairs. In some cases, using an assistant from a different architecture even outperforms one from the same series under certain compression ratios.
>
> We attribute this to the assistant’s key role in estimating frame importance. As long as this estimation is accurate, architectural alignment is not essential. Additionally, when the models share similar architectures or training recipes, aligned attention distributions may further boost performance.
>
> &nbsp;
> > **Response to Weakness 3 & Question 2 (More Experimental Settings)**
>
> **(1) Inference speed-up with fixed input frames**
>
> Using LLaVA-OneVision (7B) on VideoMME with 32-frame inputs, we have evaluated DyToK across retention ratios from 10% to 75%. All statistics include both the assistant and primary models. Results show substantial GPU memory and latency reductions with minimal accuracy loss. At 50% retention, DyToK achieves 2.1× speed-up, 22.1% memory savings, and 98.5% accuracy. Even at an aggressive 10% retention, it retains 94.3% accuracy with 4.5× speed-up.
>
> |Method|Retention Ratio|FLOPs (T)|GPU Mem (GB)|Latency (ms)|VideoMME|LongVideoBench|MLVU|Avg.|
> |---|---|---|---|---|---|---|---|---|
> |Vanilla|100%|40.8|22.2|48.3|60.7|57.7|44.7|54.4|
> |DyToK|75%|32.6|20.2|44.1|59.3|57.0|43.1|53.1|
> |DyToK|50%|21.9|17.3|23.5|60.2|56.8|43.8|53.6|
> |DyToK|25%|12.2|17.3|11.3|59.9|55.3|44.5|53.2|
> |DyToK|20%|9.6|17.3|10.7|58.4|55.4|43.7|52.5|
> |DyToK|15%|8.4|17.3|10.4|58.4|54.1|42.5|51.7|
> |DyToK|10%|5.9|17.3|10.8|57.1|53.1|43.6|51.3|
>
> *Please note: The extent of memory savings largely depends on the model size.*
>
> **(2) Performance gains under fixed computational budgets**
>
> We evaluate DyToK’s capability to support longer video sequences under the same token budget as the uncompressed baseline, using Qwen2.5-VL (7B) on MLVU (token budget alignment details in Appendix H.2).
>
> *Token Budget Alignment*
> |Method|Retention Ratio|#Frames|MLVU|
> |---|---|---|---|
> |Vanilla|100%|32|44.7|
> |DyToK|57.1%|56|45.6|
> |DyToK|50%|**64**|**47.9**|
>
> We attribute the accuracy gains to DyToK’s token compression, which reduces tokens per frame and enables more frames under a fixed token budget, improving temporal granularity.
>
> To address efficiency concerns from the assistant model, we also evaluate DyToK under a fixed FLOPs budget matching the uncompressed baseline, ensuring fair computational comparison. As Qwen2.5-VL yields variable token counts due to dynamic resolution, we use LLaVA-OneVision (7B) for consistent FLOPs estimation.
>
> *FLOPs Budget Alignment*
> |Method|Retention Ratio|#Frames|MLVU|
> |---|---|---|---|
> |Vanilla|100%|64|48.0|
> |DyToK|57.1%|96|48.8|
> |DyToK|49.5%|**106**|**49.5**|
>
> &nbsp;
> > **Response to Question 1(a) & Question 3 (More Visualization)**
>
> We appreciate your constructive feedback regarding visualization. Due to rebuttal format limits, we cannot include visuals now but will add the following in the final version:
>
> - Attention distribution across different assistant model architectures
> - Performance trends across varying token retention ratios
> - Selected keyframe visualizations moved from Appendix (Figs. 7-14) to the main text
> - Comparative frame weight distributions from all query tokens vs. only the last query token
> - Attention distributions for general description queries
>
> Please feel free to suggest additional visualizations.
>
> &nbsp;
> > **Response to Question 4(a) (Additional Analysis on Textual Token Selection)**
>
> We agree that relying solely on the last textual token may not fully capture the sequence’s semantics. To explore this, we conduct ablations on multiple long video benchmarks using LLaVA-OneVision (7B) under various retention ratios.
>
> |Method|Retention Ratio|VideoMME|LongVideoBench|MLVU|Avg.|Relative Avg. (%)|
> |---|---|---|---|---|---|---|
> |Vanilla|N/A|58.5|56.6|47.1|54.1|100|
> |FastV|75%|57.6|57.1|46.5|53.7|99.3|
> |DyToK (all query tokens)|75%|58.1|**57.1**|45.4|53.5|98.9|
> |DyTok (last query token)|75%|**58.4**|56.8|**46.8**|**54.0**|**99.8**|
> |FastV|50%|57.2|57.1|44.7|53.0|98.0|
> |DyToK (all query tokens)|50%|58.2|**58.0**|45.6|53.9|99.6|
> |DyTok (last query token)|50%|**58.5**|57.2|**46.3**|**54.0**|**99.8**|
> |FastV|25%|56.0|56.6|43.7|52.1|96.3|
> |DyToK (all query tokens)|25%|**56.9**|**58.0**|47.3|**54.1**|**100**|
> |DyTok (last query token)|25%|56.3|55.7|**47.8**|53.3|98.5|
> |FastV|15%|51.1|51.2|38.3|46.9|86.7|
> |DyToK (all query tokens)|15%|54.1|**53.1**|**44.4**|**50.5**|**93.3**|
> |DyTok (last query token)|15%|**54.8**|52.6|43.2|50.2|92.8|
>
> As shown above, at higher retention ratios (e.g., 75%, 50%), using only the last query token performs better. At more aggressive ratios (e.g., 25%, 15%), using all query tokens yields further gains. We hypothesize that the last token acts as a precise but narrow selector, while all tokens offer broader coverage with higher recall. When visual evidence is abundant, precision reduces noise; when it is scarce, broader semantic coverage becomes essential.
>
> &nbsp;
> > **Response to Question 4(b) (Concern about General Descriptive Queries)**
>
> **Quantitative Analysis.** To assess DyToK’s ability to select keyframes for general descriptive queries, we conduct captioning experiments on VideoChatGPT using LLaVA-OneVision (7B), which includes prompts like “What is happening in the video?” Following Dynamic-VLM[1], we use GPT-3.5-Turbo-0613 for quantitative scoring.
>
> |Method|Retention Ratio|CO|CU|CI|DO|TU|Avg.|Relative Avg. (%)|
> |---|---|---|---|---|---|---|---|---|
> |Vanilla|N/A|3.11|3.17|2.79|2.83|2.20|2.82|100.0|
> |VisionZip|50%|2.91|3.08|2.72|2.74|2.03|2.70|95.7|
> |DyToK|50%|**3.04**|**3.15**|**2.79**|**2.80**|**2.17**|**2.79**|**98.9**|
> |VisionZip|25%|2.62|2.86|2.50|2.42|1.81|2.44|86.5|
> |DyToK|25%|**3.01**|**3.11**|**2.76**|**2.68**|**2.10**|**2.73**|**96.8**|
> |VisionZip|15%|2.20|2.48|2.14|2.08|1.61|2.10|74.5|
> |DyToK|15%|**2.91**|**3.02**|**2.67**|**2.54**|**1.97**|**2.62**|**92.9**|
> |VisionZip|10%|1.46|1.85|1.51|1.58|1.24|1.53|54.3|
> |DyToK|10%|**2.62**|**2.75**|**2.38**|**2.24**|**1.84**|**2.37**|**84.0**|
>
> **Qualitative Analysis.** In visualization experiments, we have observed that DyToK effectively identifies both “semantic keyframes” (relevant to query semantics) and “content keyframes” (capturing key scene or event transitions), making it well-suited for general descriptive queries. This demonstrates an advantage over CLIP-based methods that rely solely on semantic matching. While visualizations cannot be included due to rebuttal constraints, they will appear in the final version along with detailed results.
>
> To support our claim during the rebuttal phase, we replace DyToK’s cross-modal attention-based frame weight estimation with CLIP-based semantic similarity. Implementation details follow AKS[2]. Results remain consistent.
>
> |Method|Retention Ratio|Short|Medium|Long|Overall|Relative Avg. (%)|
> |---|---|---|---|---|---|---|
> |Vanilla|N/A|69.9|56.7|48.9|58.5|100|
> |DyToK (w/ CLIP)|25%|68.3|**56.9**|**49.1**|58.1|99.4|
> |DyToK|25%|**69.2**|56.8|48.9|**58.3**|**99.7**|
> |DyToK (w/ CLIP)|15%|64.6|53.9|48.4|55.6|95.1|
> |DyToK|15%|**66.3**|**55.0**|**48.7**|**56.7**|**97.0**|
> |DyToK (w/ CLIP)|10%|57.6|50.9|45.8|51.4|87.9|
> |DyToK|10%|**61.0**|**51.1**|**47.6**|**53.2**|**91.0**|
>
> [1] Dynamic-VLM: Simple Dynamic Visual Token Compression for VideoLLM
>
> [2] Adaptive Keyframe Sampling for Long Video Understanding

---

> > ### Comment · Reviewer_bZfF · 2025-08-07
> >
> > Thank you for the thoughtful and comprehensive rebuttal. Most of my concerns have been reasonably addressed. The final performances provided by the authors demonstrate that guidance from assistant models, whether from the same or different architecture, can be effective across several tested models.
> >
> > However, I still retain one concern: whether the effectiveness of the proposed method arises primarily from empirical observations on the specific models studied, or if it is supported by deeper insights into the underlying relationships between attention patterns across different architectures. As noted in Review Question 1, a more direct statistical analysis of attention distribution alignment and its correlation with final performance would make the method more convincing.

---

> > > ### Author Response · Authors · 2025-08-08
> > >
> > > Thank you for your valuable and constructive comments. We agree that a direct statistical analysis can provide a more intuitive illustration of the insight underlying DyToK. Therefore, we select two architectures with notable differences in the open-source community, i.e., LLaVA-OneVision (OV) and Qwen2.5-VL (Qwen), as base models. Then, with the VideoMME dataset, we examine the performance (Acc) with the retention ratios of 25%, 20%, and 15% of these two models to investigate the underlying relationships between attention patterns across architectures, as well as the correlation between attention distribution and final performance.
> > >
> > > Specifically, in our experiments, we quantify the relationships between attention distributions from the following two complementary perspectives:
> > >
> > > - **Global attention patterns:** We compute the **KL divergence** between the attention weights of the assistant and primary models to characterize the overall alignment of their attention distributions. The smaller the KL divergence is, the better the two distributions are globally aligned.
> > > - **Local keyframe perception:** We calculate the **overlap ratio of the top-K most attended frames** between the assistant and primary models to measure similarity in the positions of key frames attended to. The higher top-K overlap ratio indicates a more similar local pattern.
> > >
> > > | Assistant/Primary Model Pairs | KL Divergence | Top-8 Overlap (%) | Top-16 Overlap (%) | Acc (25%) | Acc (20%) | Acc (15%) |
> > > | --- | --- | --- | --- | --- | --- | --- |
> > > | OV (0.5B) guides OV (7B) | 0.06 | 68.1 | 76.9 | 58.3 | 57.8 | 56.7 |
> > > | Qwen (3B) guides OV (7B) | 0.11 | 64.8 | 75.4 | 58.9 | 57.6 | 56.6 |
> > > | Qwen (3B) guides Qwen (7B) | 0.02 | 75.0 | 81.3 | 60.6 | 60.0 | 58.3 |
> > > | OV (0.5B) guides Qwen (7B) | 0.19 | 64.3 | 74.9 | 60.5 | 59.7 | 57.3 |
> > >
> > > These results lead to the following observations:
> > >
> > > - **Correlation of attention distributions across architectures:**
> > >     - *Global attention patterns differ substantially.* Owing to differences in architecture and training recipes, KL divergence across architectures is generally higher than within the same architecture. This gap limits direct cross-architecture enhancement for token pruning, thus necessitating the need for DyToK that leverages local keyframe perception for accelerating VLLMs.
> > >     - *Local keyframe perception remains highly consistent.* Regardless of architecture, top-K overlaps remain high, indicating that the keyframe prior is an inherent property of VLLMs and is architecture-independent. This observation can help explain why DyTok can generalize across different architectures.
> > > - **Global attention patterns have little correlation with performance.** Regardless of differences in global patterns, DyToK consistently yields decent and comparable performance. This supports our rebuttal statement that the assistant’s key role is to estimate frame importance, and as long as the keyframe prior estimation is generally accurate, the global attention similarity between the assistant and primary models is not necessary for ensuring the final performance.
> > > - **Local keyframe perception is positively correlated with performance.** Across different architectures and retention ratios, higher top-K overlap consistently indicates better final performance. This observation suggests that DyToK’s efficacy primarily relies on local keyframe perception rather than global attention patterns, thereby enabling cross-architecture applications.
> > >
> > > Besides, to further investigate the **impact of model size**, we conduct experiments **across different model parameters** within the Qwen2.5-VL family (3B, 7B, and 32B).
> > >
> > > | Assistant/Primary Model Pairs | KL Divergence | Top-8 Overlap (%) | Top-16 Overlap (%) | Acc (25%) | Acc (20%) | Acc (15%) |
> > > | --- | --- | --- | --- | --- | --- | --- |
> > > | Qwen (3B) guides Qwen (7B) | 0.02 | 75.0 | 81.3 | 60.6 | 60.0 | 58.3 |
> > > | Qwen (3B) guides Qwen (32B) | 0.18 | 70.0 | 76.7 | 58.5 | 57.9 | 54.7 |
> > > | Qwen (7B) guides Qwen (32B) | 0.17 | 73.8 | 77.5 | 59.2 | 58.1 | 55.8 |
> > >
> > > *Note: Due to computational constraints, Qwen2.5-VL (32B) is run with 4-bit quantization.*
> > >
> > > The results indicate that even within the same family, global attention patterns can differ considerably. In contrast, local keyframe perception retains high similarity and is positively correlated with accuracy.
> > >
> > > In summary, we hope the above statistical analysis can help explain the DyToK’s applicability to diverse assistant–primary model pairs with different architectures and parameter scales. We sincerely appreciate your constructive feedback, and we will incorporate these analyses into the final version to provide a clearer and more data-driven presentation of DyToK’s core insight.

---

> > > > ### Comment · Reviewer_bZfF · 2025-08-09
> > > >
> > > > Thanks for the detailed responses. I suggest the author to include these intuitive analyses in the final version. Overall, my concerns have mostly been resolved. I'll update the score accordingly.

---

> > > > > ### Author Response · Authors · 2025-08-09
> > > > >
> > > > > Thank you for your positive feedback and insightful review. We will incorporate your suggestions and these analyses into the final version to further strengthen our work. We sincerely expect and appreciate your support.

---

> > > ### Author Response · Authors · 2025-08-09
> > >
> > > Thank you for your thoughtful feedback. As the discussion period will conclude in less than eight hours, we would be very happy to provide any further clarification if needed. We sincerely appreciate your time and reconsideration of our work.

---

> ### Author Response · Authors · 2025-08-06
> **Look Forward to Feedback from Reviewer bZfF**
>
> Dear Reviewer bZfF,
>
> We sincerely appreciate the time and effort you devoted to reviewing our manuscript. In response to your thoughtful feedback, we have submitted a rebuttal with extensive experimental results and clarifications addressing your concerns, which includes the following key points:
>
> - **Cross-Architecture Transferability.** Extensive experiments show that DyToK remains robust with cross-architecture assistant–primary pairs, sometimes even outperforming same-series models under certain compression ratios.
> - **Fixed Input Frames / Computational Budget.** We assess DyToK’s efficiency (latency, FLOPs, memory) with 32-frame input, and evaluate its performance gains under fixed token and FLOPs budgets.
> - **Analysis on Textual Token Selection.** Ablations show that the last query token excels at higher retention, while all tokens perform better under aggressive compression, suggesting a precision–recall trade-off.
> - **Analysis on General Descriptive Queries.** DyToK effectively identifies both semantic and content keyframes, outperforming semantic matching in qualitative and quantitative evaluations, with CLIP-based ablations further supporting the results.
>
> We hope that these clarifications and additional experiments can address your concerns. Thank you again for your constructive review, and we welcome any further feedback or questions.
>
> Sincerely,
>
> Authors of Paper #19575

---

### Official Review · Reviewer_mobj · 2025-07-01

**Clarity:** 3
**Significance:** 2
**Originality:** 2
**Rating:** 4
**Confidence:** 4

**Summary:**

This paper introduces DyTok , a training-free dynamic token compression method that leverages LLM-guided keyframe priors. Given a sequence of visual tokens, DyTok employs a LLM to assess the importance of each token by computing its similarity with the textual query tokens. Based on these importance scores, the method selectively retains the most informative tokens while maintaining a balanced token distribution across frames. DyTok is evaluated on multiple video understanding benchmarks.

**Questions:**

Please refer to the weaknesses part for my concerns. My initial judgment is BR, and I will adjust my rating accordingly based on the authors' response.

**Ethical Concerns:**

["NO or VERY MINOR ethics concerns only"]

**Final Justification:**

My initial judgment is on the borderline. The authors' rebuttal addressed my concerns, especially the transferability of the layer selection. Although I'm still on the borderline, I lean towards to the positive side, and my final rating is BA.

**Limitations:**

yes

**Quality:**

3

**Strengths And Weaknesses:**

Strengths:
* The paper is well-written and clearly structured, making it easy to follow.
* The proposed idea is conceptually straightforward, computationally efficient, and can be integrated into various methods.
* Extensive experimental results demonstrate the effectiveness and robustness of DyTok across multiple tasks and datasets.

Weaknesses:
* The paper computes attention scores between the final textual query token and the visual tokens. However, it would be helpful to include an analysis on the choice of textual token representation. While the last token may capture global query information, in some cases, it might lack sufficient semantic richness.
* Although the auxiliary lightweight assistant model helps reduce computational cost, it still relies on strong pre-trained VLMs, such as LLaVA-OneVision and Qwen2.5-VL. This dependency may limit the generalizability and accessibility of DyTok across different frameworks or less powerful backbones.
* The selection of deep layers appears to be based on specific architectural choices. It is unclear whether this selection is model-agnostic or tailored to particular backbones. If it is not, then the analysis presented in Table 3 may need to be repeated for each new model architecture. Is there evidence supporting the transferability of this layer selection strategy?
* In the appendix Table 7, instead of only computing prefilling time, what about the inference time of the whole pipeline? As an additional model is used, I'd expect the DyTok pipeline to have longer inference time overall.

---

> ### Author Rebuttal · Authors · 2025-07-31
>
> We sincerely appreciate your comments. All suggestions will be incorporated into the paper. Our responses are as follows.
>
> &nbsp;
>
> > **Response to Weakness 1 (Additional Analysis on Textual Token Selection)**
>
> Thank you for the insightful suggestion. We agree that relying solely on the last textual token may not fully capture the sequence’s semantics. To explore this, we conduct ablations on multiple long video benchmarks using LLaVA-OneVision (7B) under various retention ratios.
>
> |Method|Retention Ratio|VideoMME|LongVideoBench|MLVU|Avg.|Relative Avg. (%)|
> |---|---|---|---|---|---|---|
> |Vanilla|N/A|58.5|56.6|47.1|54.1|100|
> |FastV|75%|57.6|57.1|46.5|53.7|99.3|
> |DyToK (all query tokens)|75%|58.1|**57.1**|45.4|53.5|98.9|
> |DyTok (last query token)|75%|**58.4**|56.8|**46.8**|**54.0**|**99.8**|
> |FastV|50%|57.2|57.1|44.7|53.0|98.0|
> |DyToK (all query tokens)|50%|58.2|**58.0**|45.6|53.9|99.6|
> |DyToK (last query token)|50%|**58.5**|57.2|**46.3**|**54.0**|**99.8**|
> |FastV|25%|56.0|56.6|43.7|52.1|96.3|
> |DyToK (all query tokens)|25%|**56.9**|**58.0**|47.3|**54.1**|**100**|
> |DyTok (last query token)|25%|56.3|55.7|**47.8**|53.3|98.5|
> |FastV|15%|51.1|51.2|38.3|46.9|86.7|
> |DyToK (all query tokens)|15%|54.1|**53.1**|**44.4**|**50.5**|**93.3**|
> |DyTok (last query token)|15%|**54.8**|52.6|43.2|50.2|92.8|
>
>
>
> As shown above, at higher retention ratios (e.g., 75%, 50%), using only the last query token performs better. At more aggressive ratios (e.g., 25%, 15%), using all query tokens yields further gains. We hypothesize that the last token acts as a precise but narrow selector, while all tokens offer broader coverage with higher recall. When visual evidence is abundant, precision reduces noise; when it is scarce, broader semantic coverage becomes essential.
>
> We will incorporate these analyses into the final version and add visualizations comparing the distributions of frame weights derived from both strategies.
>
> &nbsp;
>
> > **Response to Weakness 2 (Concern on Cross-Architecture Transferability and Weaker Backbone Adaptability)**
>
> We sincerely appreciate your insightful suggestion. We agree that validating DyToK's transferability across different model architectures is crucial to demonstrate its generalizability and accessibility. To address this, we conduct experiments on VideoMME, using assistant and primary models with distinctly different architectures under varying retention ratios.
>
> To maximize the differences in architectures and thus rigorously test DyToK's robustness, we select LLaVA-OneVision (OV) and Qwen2.5-VL (Qwen), two open-source frameworks exhibiting significant architectural differences. We believe that strong performance under such settings provides meaningful evidence of generalizability.
>
> Preliminary results are shown below, with OV using 0.5B/7B and Qwen using 3B/7B as assistant and primary models.
>
> |Method|Retention Ratio|Short|Medium|Long|Overall|Relative Acc. (%)|
> |---|---|---|---|---|---|---|
> |Vanilla (OV)|N/A|69.9|56.7|48.9|58.5|100.0|
> |VisionZip|75%|68.9|56.2|48.2|57.8|98.8|
> |DyToK (OV guides OV)|75%|70.8|**57.2**|**48.9**|**59.0**|**100.9**|
> |DyToK (Qwen guides OV)|75%|**71.4**|57.0|48.3|58.9|100.7|
> |VisionZip|50%|66.8|56.6|48.2|57.2|97.8|
> |DyToK (OV guides OV)|50%|**71.9**|56.1|**49.3**|**59.1**|**101.0**|
> |DyToK (Qwen guides OV)|50%|71.6|**56.6**|48.1|58.7|100.3|
> |VisionZip|25%|62.2|53.0|47.4|54.2|92.6|
> |DyToK (OV guides OV)|25%|69.2|**56.8**|**48.9**|58.3|99.7|
> |DyToK (Qwen guides OV)|25%|**71.3**|56.6|48.1|**58.9**|**100.7**|
> |VisionZip|20%|60.2|51.0|45.7|52.3|89.4|
> |DyToK (OV guides OV)|20%|**69.2**|**56.4**|47.8|**57.8**|**98.8**|
> |DyToK (Qwen guides OV)|20%|68.7|55.9|**48.2**|57.6|98.5|
> |VisionZip|15%|55.9|49.9|44.3|50.0|85.5|
> |DyToK (OV guides OV)|15%|66.3|**55.0**|48.7|**56.7**|**96.9**|
> |DyToK (Qwen guides OV)|15%|**67.3**|53.7|**48.9**|56.6|96.8|
> |VisionZip|10%|47.0|44.2|42.2|44.5|76.1|
> |DyToK (OV guides OV)|10%|**61.0**|51.1|47.6|53.2|90.9|
> |DyToK (Qwen guides OV)|10%|59.2|**53.3**|**47.9**|**53.5**|**91.5**|
>
> |Method|Retention Ratio|Short|Medium|Long|Overall|Relative Acc. (%)|
> |---|---|---|---|---|---|---|
> |Vanilla (Qwen)|N/A|72.6|61.4|49.9|61.3|100.0|
> |FastV|75%|70.0|57.4|50.4|59.3|96.7|
> |DyToK (Qwen guides Qwen)|75%|72.6|**60.6**|**51.0**|**61.4**|**100.1**|
> |DyToK (OV guides Qwen)|75%|**72.7**|60.1|50.2|61.0|99.5|
> |FastV|50%|63.9|52.7|48.0|54.9|89.5|
> |DyToK (Qwen guides Qwen)|50%|**72.0**|60.2|**51.7**|**61.3**|**100.0**|
> |DyToK (OV guides Qwen)|50%|71.8|**60.4**|51.0|61.1|99.6|
> |FastV|25%|59.9|49.1|46.8|51.9|84.7|
> |DyToK (Qwen guides Qwen)|25%|**71.7**|**58.9**|51.3|**60.6**|**98.9**|
> |DyToK (OV guides Qwen)|25%|71.3|58.3|**51.8**|60.5|98.7|
> |FastV|20%|57.8|48.4|45.6|50.6|82.5|
> |DyToK (Qwen guides Qwen)|20%|**70.8**|**57.7**|**51.4**|**60.0**|**97.8**|
> |DyToK (OV guides Qwen)|20%|**70.8**|57.2|51.2|59.7|97.5|
> |FastV|15%|54.9|45.0|44.2|48.0|78.4|
> |DyToK (Qwen guides Qwen)|15%|**68.4**|**56.3**|**50.1**|**58.3**|**95.1**|
> |DyToK (OV guides Qwen)|15%|67.2|55.1|49.7|57.3|93.5|
>
> The results show that DyToK remains robust even with cross-architecture assistant-primary model pairs. In some cases, using an assistant from a different architecture even outperforms one from the same series under certain compression ratios.
>
> We attribute this to the assistant’s key role in estimating frame importance. As long as this estimation is accurate, architectural alignment is not essential. Additionally, when the models share similar architectures or training recipes, aligned attention distributions may further boost performance.
>
> These comprehensive experimental findings broaden DyToK's practical applicability in various scenarios:
>
> - **No lightweight version available**: A lightweight variant from another model, such as LLaVA-OneVision 0.5B, can serve effectively as an assistant model, given its proven efficacy and extremely small parameter count.
> - **Inadequate capacity of lightweight models**: If a lightweight model lacks sufficient capability to generate accurate frame weights, a lightweight variant from another pre-trained video large language model can readily serve as the assistant model.
>
> We hope this addresses your concerns regarding DyToK’s cross-architecture transferability and robustness with weaker backbones. Given the importance of this issue to DyToK’s versatility, we will elaborate on this in the main text and add visualizations in the appendix to illustrate attention distribution differences across assistant architectures.
>
> &nbsp;
>
> > **Response to Weakness 3 (Additional Analysis on Deep Layer Selection)**
>
> As stated in Section 3.2 (line 154), we define deep layers as the last 25% of the assistant model’s layers to ensure generalizability without architecture-specific tuning. This proportional strategy adapts well across models like LLaVA-OneVision and Qwen2.5-VL, making it unnecessary to repeat Table 3 experiments for each new model.
>
> The ablation in Table 3 serves to validate our key observation from Section 2.2 (line 113) that “deep attention layers provide good keyframe priors.” Appendix A.4 (Table 5) further analyzes individual layers and combinations divided into quarters and sixths, confirming the benefits of aggregating deep layers.
>
> These results support, but do not affect, our practical choice of using the last 25% layers. We appreciate the reviewer’s suggestion and will clarify the distinction between our empirical analyses and adopted strategy in Section 3.2 to avoid misunderstanding.
>
> To further illustrate the transferability of our approach, we also conduct additional layer analyses on LLaVA-OneVision and Qwen2.5-VL.
>
> *LLaVA-OneVision*
>
> ||VideoMME|LongVideoBench|MLVU|Avg.|
> |---|---|---|---|---|
> |All Layers|57.1|54.3|44.7|52.0|
> |Shallow 25%|55.5|53.6|43.7|50.9|
> |Mid-Shallow 25%|56.4|54.1|43.7|51.4|
> |Mid-Deep 25%|57.5|55.1|42.7|51.8|
> |Deep 25%|**57.8**|**56.0**|**45.1**|**53.0**|
>
> *Qwen2.5-VL*
>
> ||VideoMME|LongVideoBench|MLVU|Avg.|
> |---|---|---|---|---|
> |All Layers|58.0|54.3|41.9|51.4|
> |Shallow 25%|57.6|54.2|42.1|51.3|
> |Mid-Shallow 25%|57.9|54.0|**42.6**|51.5|
> |Mid-Deep 25%|58.7|54.9|42.2|51.9|
> |Deep 25%|**59.0**|**55.8**|41.7|**52.2**|
>
> The experimental results demonstrate that our layer selection strategy is model- and architecture-agnostic, exhibiting generalizability by eliminating the need for repeated layer analyses when adapting to new models.
>
> &nbsp;
>
> > **Response to Weakness 4 (Inference Time Clarification)**
>
> The “prefilling time” reported in Appendix Table 7 reflects the combined inference time of the assistant and primary models during the generation of the first token in DyToK. We focus on this metric because DyToK’s additional computation occurs entirely during prefilling. In contrast, decoding involves only the primary model operating on pruned tokens, leading to consistent speedups. Measuring total generation time would understate DyToK’s actual overhead, as the assistant’s cost becomes negligible over longer sequences. Therefore, prefilling time offers a fair and transparent efficiency measure.
>
> We further validate this choice through additional experiments on long-sequence captioning with VideoChatGPT, which confirm our analysis.
>
> |Method|Retention Ratio|Latency (s)|CO|CU|CI|DO|TU|Avg.|
> |---|---|---|---|---|---|---|---|---|
> |Vanilla|100%|1.41|3.11|3.17|2.79|2.83|2.20|2.82|
> |DyToK|75%|1.39|3.05|3.14|2.76|2.80|2.20|2.79|
> |DyToK|50%|1.07|3.04|3.15|2.79|2.80|2.17|2.79|
> |DyToK|25%|0.66|3.01|3.11|2.76|2.68|2.10|2.73|
> |DyToK|20%|0.57|2.96|3.01|2.71|2.60|2.00|2.67|
> |DyToK|15%|0.54|2.91|3.02|2.67|2.54|1.97|2.62|
> |DyToK|10%|0.27|2.62|2.75|2.38|2.24|1.84|2.37|

---

> > ### Comment · Reviewer_mobj · 2025-08-05
> >
> > Thanks for the detailed response. The new results further validate the claims made in the papers, and my concerns are mostly well addressed. I'll update the score accordingly.

---

> ### Author Response · Authors · 2025-08-05
>
> We sincerely appreciate your positive feedback and constructive review. If you have any additional questions, we look forward to further discussions.

---

### Official Review · Reviewer_nbsC · 2025-07-02

**Clarity:** 4
**Significance:** 3
**Originality:** 3
**Rating:** 4
**Confidence:** 4

**Summary:**

The paper introduces DyToK, a training-free dynamic token compression framework for Video Large Language Models (VLLMs). By extracting query-conditioned keyframe priors from deep attention layers of a lightweight assistant model, DyToK adaptively allocates per‐frame token budgets and applies any compatible pruning strategy. This plug-and-play approach seamlessly integrates with existing compression backbones—both encoder feature– and attention–based—achieving state-of-the-art efficiency-accuracy trade-offs across multiple long-video benchmarks.

**Questions:**

Does the reported efficiency improvement include the lightweight assistant model’s forward pass?

**Ethical Concerns:**

["NO or VERY MINOR ethics concerns only"]

**Limitations:**

yes

**Quality:**

3

**Strengths And Weaknesses:**

Strengths:
* The motivation for dynamic compression is presented concisely, highlighting inefficiencies of uniform frame pruning.
* The ablation studies systematically evaluate compression ratios, attention‐layer selection, and assistant model size, providing strong empirical insights.
* Algorithm 1 is presented with pseudocode that precisely describes the adaptive token allocation and compression steps.
* DyToK’s training-free paradigm and modular compression function enable straightforward integration with existing token-pruning methods.
* The paper demonstrates insightful finding that depth of attention is more critical than model scale for importance estimation.
DyToK reliably boosts accuracy under aggressive pruning across diverse benchmarks and methods.

Weaknesses:
*  While existing binary keyframe sampling of VLLM is discussed, the method also closely relates to the retrieval-based video understanding methods, such as rag-based methods(Video-rag: Visually-aligned retrieval-augmented long video comprehension). It would add clarity if the authors discuss and compare with that line of methods.
*  Eq. (1) quantifies total FLOPs, but the paper does not present calculated FLOP savings for DyToK relative to baselines.
*  It remains unclear how different assistant models (a wider ranges besides 7B and 0.5B) influence the performance, and how mispredictions of assistant models propagate to the primary VLLM.

---

> ### Author Rebuttal · Authors · 2025-07-31
>
> We sincerely appreciate your comments. All suggestions will be incorporated into the paper. Our responses are as follows.
>
> &nbsp;
>
> > **Response to Weakness 1 (More Discussion with RAG-Based Methods)**
> >
>
> Thank you for your insightful suggestion. Video-RAG primarily focuses on enhancing video comprehension through auxiliary textual information extracted by external speech recognition, OCR, and object detection modules. In contrast, our method specifically targets efficient long-video reasoning and extended video support within a given computational budget, making a direct comparison challenging. A commonality between the two methods is that Video-RAG employs CLIP-based keyframe extraction in its object detection stage.
>
> To facilitate a clearer comparison regarding the handling of keyframes, we incorporate Video-RAG’s keyframe extraction module into DyToK’s temporal importance estimation stage. Specifically, we utilize CLIP’s cross-modal semantic similarity scores to determine frame weights, dynamically allocating token retention ratios for each frame. We present experimental results on VideoMME using LLaVA-OneVision as the base model under different retention ratios below. We will incorporate these discussions and experimental results into our paper.
>
> | Method  | Retention Ratio | Short | Medium | Long | Overall | Relative Acc. (%) |
> | --- | --- | --- | --- | --- | --- | --- |
> | Vanilla | N/A | 69.9 | 56.7 | 48.9 | 58.5 | 100 |
> | Video-RAG | 25% | 68.3 | **56.9** | **49.1** | 58.1 | 99.4 |
> | DyToK | 25% | **69.2** | 56.8 | 48.9 | **58.3** | **99.7** |
> | Video-RAG | 15% | 64.6 | 53.9 | 48.4 | 55.6 | 95.1 |
> | DyToK | 15% | **66.3** | **55.0** | **48.7** | **56.7** | **97.0** |
> | Video-RAG | 10% | 57.6 | 50.9 | 45.8 | 51.4 | 87.9 |
> | DyToK | 10% | **61.0** | **51.1** | **47.6** | **53.2** | **91.0** |
>
> &nbsp;
>
> > **Response to Weakness 2 (Relative FLOPs Saving Calculation)**
> >
>
> We appreciate your valuable feedback on presenting the relative computational savings. To intuitively demonstrate the practical efficiency gains achieved with minimal performance trade-offs, we conduct experiments on VideoMME using LLaVA-OneVision (7B) as the base model, evaluating GPU inference latency, FLOPs, and memory usage across a broad retention ratio range (10%-75%) with 32-frame input. The summarized results are presented in the following table, with detailed efficiency analysis provided in Appendix D. All statistics include both the assistant and primary models.
>
> | Method | Retention Ratio | FLOPs (T) | GPU Memory (GB) | Latency (ms) | VideoMME | LongVideoBench | MLVU | Avg. |
> | --- | --- | --- | --- | --- | --- | --- | --- | --- |
> | Vanilla | 100% | 40.8 | 22.2 | 48.3 | 60.7 | 57.7 | 44.7 | 54.4 |
> | DyToK | 75% | 32.6 | 20.2 | 44.1 | 59.3 | 57.0 | 43.1 | 53.1 |
> | DyToK | 50% | 21.9 | 17.3 | 23.5 | 60.2 | 56.8 | 43.8 | 53.6 |
> | DyToK | 25% | 12.2 | 17.3 | 11.3 | 59.9 | 55.3 | 44.5 | 53.2 |
> | DyToK | 20% | 9.6 | 17.3 | 10.7 | 58.4 | 55.4 | 43.7 | 52.5 |
> | DyToK | 15% | 8.4 | 17.3 | 10.4 | 58.4 | 54.1 | 42.5 | 51.7 |
> | DyToK | 10% | 5.9 | 17.3 | 10.8 | 57.1 | 53.1 | 43.6 | 51.3 |
>
> *Notably, the extent of memory savings significantly depends on the model size.*
>
> From the table above, DyToK achieves an excellent trade-off between accuracy and efficiency:
>
> - **GPU Memory and Inference Savings:** DyToK achieves a 2.1× inference acceleration at a 50% retention ratio, and this acceleration increases to 4.3× at a 25% retention ratio. Starting from a 50% retention ratio, GPU memory usage remains around 17.3 GB, which essentially represents the minimum memory footprint determined solely by the model weights.
> - **Accuracy-Efficiency Trade-Off:** At a moderate retention ratio of 50%, DyToK retains 98.5% of the original accuracy, reduces inference time by 2.1×, and decreases GPU memory usage by 22.1%. At a 25% retention ratio, DyToK maintains 97.8% accuracy with a 4.3× speedup. Even at the extreme retention ratio of 10% (only ~14 tokens per frame), DyToK maintains 94.3% accuracy.
>
> To more comprehensively evaluate DyToK’s efficiency, we further conduct long-sequence captioning experiments on VideoChatGPT. Following Dynamic-VLM[1], we use GPT-3.5-Turbo-0613 for quantitative scoring. All statistics include both the assistant and primary models.
>
> | Method | Retention Ratio | Latency (s) | CO | CU | CI | DO | TU | Avg. |
> | --- | --- | --- | --- | --- | --- | --- | --- | --- |
> | Vanilla | 100% | 1.41 | 3.11 | 3.17 | 2.79 | 2.83 | 2.20 | 2.82 |
> | DyToK | 75% | 1.39 | 3.05 | 3.14 | 2.76 | 2.80 | 2.20 | 2.79 |
> | DyToK | 50% | 1.07 | 3.04 | 3.15 | 2.79 | 2.80 | 2.17 | 2.79 |
> | DyToK | 25% | 0.66 | 3.01 | 3.11 | 2.76 | 2.68 | 2.10 | 2.73 |
> | DyToK | 20% | 0.57 | 2.96 | 3.01 | 2.71 | 2.60 | 2.00 | 2.67 |
> | DyToK | 15% | 0.54 | 2.91 | 3.02 | 2.67 | 2.54 | 1.97 | 2.62 |
> | DyToK | 10% | 0.27 | 2.62 | 2.75 | 2.38 | 2.24 | 1.84 | 2.37 |
>
> [1] Dynamic-VLM: Simple Dynamic Visual Token Compression for VideoLLM
>
> &nbsp;
>
> > **Response to Weakness 3(a) (Broader Model Size)**
> >
>
> Thank you for your valuable suggestion. Beyond the **0.5B** guiding **7B** experiments using LLaVA-OneVision already presented in the main text, we conduct additional extensive experiments using Qwen2.5-VL on the MLVU benchmark, which covers a wide range of commonly used model sizes (**3B**, **7B,** and **32B**), as shown below.
>
> | Method | 100% | 25% | 20% | 15% |
> | --- | --- | --- | --- | --- |
> | Vanilla | 41.4 | N/A | N/A | N/A |
> | FastV | N/A | 23.6 | 22.3 | 17.1 |
> | DyToK (3B guides 32B) | N/A | 35.5 | **35.4** | 29.9 |
> | DyToK (7B guides 32B) | N/A | **36.3** | 34.9 | **33.9** |
>
> These experimental results provide two further insights:
>
> 1. **Video models inherently possess keyframe priors regardless of model size.**
> 2. **Such priors are not constrained by model scale, as even lightweight models can exhibit keyframe perception capabilities comparable to those of larger models.** Notably, at a 20% retention ratio, the 3B Qwen2.5-VL assistant model surpasses the 7B variant by a substantial margin, further validating the feasibility of using a lightweight assistant model to provide keyframe priors in DyToK.
>
> &nbsp;
>
> > **Response to Weakness 3(b) (More Analysis on Assistant Model Misprediction Propagation)**
> >
>
> Thank you for raising this critical point. In the DyToK framework, the assistant model’s role is strictly confined to extracting frame weights based on keyframe priors, guiding dynamic token retention without directly passing predictions to the primary model.
>
> As illustrated in Figure 1 of the main text, we have empirically found that **even when the assistant model’s predictions are incorrect, its cross-modal attention remains accurately aligned with keyframes**. Consequently, the transferred frame weights remain generally reliable, reducing the risk of error propagation. This robustness underlies DyToK’s consistent effectiveness across various base models, benchmarks, and retention ratios.
>
> Following your suggestion, we will expand discussions on misprediction propagation in the main text and provide further visual analysis beyond the current visualizations in Appendix Figures 7-14.
>
> &nbsp;
>
> > **Response to Question (Efficiency Improvement Confirmation)**
> >
>
> **Yes.** To ensure fairness, efficiency gains reported in our study (Appendix D) include the forward passes of both assistant and primary models.

---

> ### Author Response · Authors · 2025-08-06
> **Look Forward to Feedback from Reviewer nbsC**
>
> Dear Reviewer nbsC,
>
> We sincerely appreciate the time and effort you devoted to reviewing our manuscript. In response to your thoughtful feedback, we have submitted a rebuttal with extensive experimental results and clarifications addressing your concerns, which includes the following key points:
>
> - **Discussion with RAG-Based Methods.** We compare DyToK with CLIP-based keyframe extraction in Video-RAG through detailed analysis and experiments.
> - **Relative FLOPs Saving Calculation.** DyToK is evaluated on VideoMME across retention ratios in terms of latency, FLOPs, and memory, and further validated on VideoChatGPT for long-sequence captioning, demonstrating strong accuracy-efficiency trade-offs.
> - **Broader Model Size.** Experiments on 3B, 7B, and 32B models reveal that video large language models inherently possess keyframe priors, regardless of model size.
> - **Analysis on Assistant Model Misprediction Propagation.** Experiments show that even with incorrect predictions, the assistant model maintains accurate cross-modal attention to keyframes, which underlies DyToK’s robustness.
>
> We hope that these clarifications and additional experiments can address your concerns. Thank you again for your constructive review, and we welcome any further feedback or questions.
>
> Sincerely,
>
> Authors of Paper #19575

---

### Official Review · Reviewer_HiJ7 · 2025-07-05

**Clarity:** 2
**Significance:** 3
**Originality:** 2
**Rating:** 3
**Confidence:** 3

**Summary:**

This paper proposes a dynamic token compression scheme that leverages the inherent attention mechanisms of VLLM. Experimental results show that the attention module effectively learns keyframe priors from the training data. By adopting this scheme, the algorithm dynamically adjusts the number of tokens on a frame-by-frame basis, adaptively removing redundant information. Extensive experiments demonstrate that DyToK achieves state-of-the-art trade-offs between efficiency and accuracy.

**Questions:**

- **Line 65: Placeholder Reference**
  The text refers to “XX datasets,” which appears to be a placeholder. This should be replaced with the actual dataset names to clearly indicate the evaluation scope and improve the paper’s professionalism.

- **Definition of Keyframe and Adaptive Rate**
  The paper does not clearly define what qualifies as a “keyframe” within the proposed method. Additionally, the mechanism for determining the adaptive token rate for keyframes versus normal frames is not sufficiently explained. A detailed description or visualization of this process would enhance understanding.

- **Runtime Efficiency**
  Although the paper emphasizes token redundancy reduction, it lacks reporting of runtime efficiency metrics. Including inference latency or throughput on real hardware would significantly strengthen the case for practical applicability, particularly in real-time or latency-sensitive environments.

**Ethical Concerns:**

["NO or VERY MINOR ethics concerns only"]

**Final Justification:**

Thank you to the authors for the detailed rebuttal. While the explanation of the attention-based importance estimation is helpful, the overall novelty of the proposed method remains limited. The lack of comparison with more recent state-of-the-art approaches—relying only on VisionZip and DyCoke as baselines—further restricts the demonstrated impact and potential for broader applicability. Given that some concerns remain unaddressed, I will maintain my rating as Borderline reject.

**Limitations:**

Yes.

**Quality:**

3

**Strengths And Weaknesses:**

### Strengths

- Extensive evaluations on long-video benchmarks convincingly demonstrate that DyToK outperforms existing state-of-the-art methods in terms of efficiency-accuracy tradeoffs.

- The paper provides a compelling finding that attention layers in vision-language large models (VLLMs) inherently encode query-conditioned keyframe priors. This insight enables effective identification of task-relevant information without requiring additional supervision.

- DyToK shows excellent generalization and compatibility across different backbone models and token compression strategies. This makes it a versatile component that can be easily integrated into existing frameworks.

- The authors conduct extensive experiments and ablation studies across various datasets and model settings, providing strong evidence for the effectiveness and robustness of the proposed approach.

### Weaknesses
- Lack of Comparison with Recent State-of-the-Art Methods: The paper omits comparison with several recently proposed token compression or pruning techniques, limiting the evaluation’s completeness and making it hard to assess the relative progress.

- No Evaluation Against Structured Compression Baselines: Methods such as LoRA, quantization-aware training, or parameter-efficient fine-tuning techniques are not included in the comparison, which raises concerns about the fairness and rigor of the benchmarks.

- Missing End-to-End Latency on Real Hardware: While theoretical FLOPs are reported, practical runtime metrics (e.g., latency on GPU inference) are not provided, making it unclear how beneficial the proposed approach is in deployment scenarios.

- Weak Justification for the Attention-Based Importance Estimation: The proposed method for selecting important tokens is intuitive but lacks theoretical grounding or sufficient ablation studies to validate why it outperforms simpler baselines like entropy- or magnitude-based filtering.

---

> ### Author Rebuttal · Authors · 2025-07-31
>
> We sincerely appreciate your comments. All suggestions will be incorporated into the paper, and typos will be fixed. Our responses are as follows.
>
> &nbsp;
>
> > **Response to Weakness 1 (Comparison with SOTA Methods)**
> >
>
> As described in Figure 2 (page 4) of the manuscript, existing token compression methods can generally be categorized into two paradigms: encoder feature-based and LLM attention-based. Accordingly, we have conducted comparisons with VisionZip[1] (state-of-the-art encoder feature-based method) and DyCoke[2] (state-of-the-art LLM attention-based method), both recently published at CVPR 2025.
>
> Under identical token budgets and across a comprehensive range of retention ratios (10%, 15%, 20%, 25%, 50%, 75%), DyToK has consistently achieved superior performance on diverse benchmarks for long video understanding (VideoMME, LongVideoBench, MLVU). Detailed comparative radar plots and complete experimental results are provided in Figure 1 (Appendix, page 2) and Tables 1 and 2 (Appendix, pages 18–19). If there are specific works that you believe we should additionally compare with, please kindly let us know, and we will do our best to include the comparisons in the discussion phase.
>
> [1] VisionZip: Longer is Better but Not Necessary in Vision Language Models (CVPR 2025)
>
> [2] DyCoke: Dynamic Compression of Tokens for Fast Video Large Language Models (CVPR 2025)
>
> &nbsp;
>
> > **Response to Weakness 2 (Evaluation Against Structured Compression Baselines)**
> >
> 1. **Training-Free vs. Training-Based:** DyToK is designed as a training-free dynamic token compression method; therefore, our evaluation primarily targets other training-free methods. As the structured compression approaches you have mentioned require training, the comparison with them is beyond the scope of our current comparative framework.
> 2. **Inference Acceleration vs. Training Acceleration:** DyToK is a training-free method that aims at accelerating inference in video large language models through dynamic token compression, whereas methods such as LoRA or parameter-efficient fine-tuning primarily target reductions in training resource consumption. Due to these distinct goals, a direct comparison is inherently challenging.
>
> &nbsp;
>
> > **Response to Weakness 3 & Question 3 (End-to-End Latency on Real Hardware)**
> >
>
> To demonstrate the real-world efficiency gains provided by our method with minimal performance compromise, we have evaluated DyToK using LLaVA-OneVision (7B) on VideoMME with 32-frame inputs. Below we summarize GPU inference latency, FLOPs, and GPU memory consumption across various token retention ratios (10% to 75%). More comprehensive efficiency analyses and experimental results can be found in Appendix D.
>
> | Method | Retention Ratio | FLOPs (T) | GPU Memory (GB) | Latency (ms) | VideoMME | LongVideoBench | MLVU | Avg. |
> | --- | --- | --- | --- | --- | --- | --- | --- | --- |
> | Vanilla | 100% | 40.8 | 22.2 | 48.3 | 60.7 | 57.7 | 44.7 | 54.4 |
> | DyToK | 75% | 32.6 | 20.2 | 44.1 | 59.3 | 57.0 | 43.1 | 53.1 |
> | DyToK | 50% | 21.9 | 17.3 | 23.5 | 60.2 | 56.8 | 43.8 | 53.6 |
> | DyToK | 25% | 12.2 | 17.3 | 11.3 | 59.9 | 55.3 | 44.5 | 53.2 |
> | DyToK | 20% | 9.6 | 17.3 | 10.7 | 58.4 | 55.4 | 43.7 | 52.5 |
> | DyToK | 15% | 8.4 | 17.3 | 10.4 | 58.4 | 54.1 | 42.5 | 51.7 |
> | DyToK | 10% | 5.9 | 17.3 | 10.8 | 57.1 | 53.1 | 43.6 | 51.3 |
>
> *Note: The extent of memory savings largely depends on model size.*
>
> As demonstrated, DyToK provides an excellent trade-off between accuracy and efficiency:
>
> - **GPU Memory and Inference Savings:** DyToK achieves a 2.1× inference acceleration at a 50% retention ratio, and this acceleration increases to 4.3× at a 25% retention ratio. Starting from a 50% retention ratio, GPU memory usage remains around 17.3 GB, which essentially represents the minimum memory footprint determined solely by the model weights.
> - **Accuracy-Efficiency Trade-Off:** At a moderate retention ratio of 50%, DyToK retains 98.5% of the original accuracy, reduces inference time by 2.1×, and decreases GPU memory usage by 22.1%. At a 25% retention ratio, DyToK maintains 97.8% accuracy with a 4.3× speedup. Even at the extreme retention ratio of 10% (only ~14 tokens per frame), DyToK maintains 94.3% accuracy.
>
> To more comprehensively evaluate DyToK’s efficiency, we further conduct long-sequence captioning experiments on VideoChatGPT. Following Dynamic-VLM[1], we use GPT-3.5-Turbo-0613 for quantitative scoring. All statistics include both the assistant and primary models.
>
> | Method | Retention Ratio | Latency (s) | CO | CU | CI | DO | TU | Avg. |
> | --- | --- | --- | --- | --- | --- | --- | --- | --- |
> | Vanilla | 100% | 1.41 | 3.11 | 3.17 | 2.79 | 2.83 | 2.20 | 2.82 |
> | DyToK | 75% | 1.39 | 3.05 | 3.14 | 2.76 | 2.80 | 2.20 | 2.79 |
> | DyToK | 50% | 1.07 | 3.04 | 3.15 | 2.79 | 2.80 | 2.17 | 2.79 |
> | DyToK | 25% | 0.66 | 3.01 | 3.11 | 2.76 | 2.68 | 2.10 | 2.73 |
> | DyToK | 20% | 0.57 | 2.96 | 3.01 | 2.71 | 2.60 | 2.00 | 2.67 |
> | DyToK | 15% | 0.54 | 2.91 | 3.02 | 2.67 | 2.54 | 1.97 | 2.62 |
> | DyToK | 10% | 0.27 | 2.62 | 2.75 | 2.38 | 2.24 | 1.84 | 2.37 |
>
> [1] Dynamic-VLM: Simple Dynamic Visual Token Compression for VideoLLM
>
> &nbsp;
>
> > **Response to Weakness 4 (More Justification for the Attention-Based Importance Estimation)**
> >
>
> Thank you for highlighting this issue. We fully agree that different importance estimation strategies warrant further investigation. Following your suggestions, we conduct additional ablation studies by comparing our cross-modal attention-based frame weighting method (Temporal Importance Estimation stage) with entropy-based and magnitude-based approaches:
>
> - **Attention Entropy-Based:** For each frame, compute the attention weights of the last query token over all visual tokens. Calculate the entropy of these attention weights and normalize the results to obtain the frame weights.
> - **Feature Entropy-Based:** Compute the information entropy of visual tokens along the feature dimension. For each frame, average the entropies of all tokens and normalize the values to derive the frame weights.
> - **Feature Magnitude-Based:** Calculate the L2 norm of visual tokens for each frame. Average the norms and normalize the results to obtain the frame weights.
>
> | Method | Retention Ratio | VideoMME | LongVideoBench | MLVU | Avg. | Relative Avg. (%) |
> | --- | --- | --- | --- | --- | --- | --- |
> | Vanilla | N/A | 58.5 | 56.6 | 47.3 | 54.1 | 100.0 |
> | Attention Entropy-Based | 25% | 58.3 | 55.9 | 45.6 | 53.3 | 98.5 |
> | Feature Entropy-Based | 25% | 58.3 | 54.8 | 42.9 | 52.0 | 96.1 |
> | Feature Magnitude-Based | 25% | 58.3 | **56.0** | 44.8 | 53.1 | 98.1 |
> | DyToK | 25% | **58.3** | 55.4 | **46.3** | **53.3** | **98.5** |
> | Attention Entropy-Based | 15% | 55.2 | 52.8 | 43.4 | 50.5 | 93.2 |
> | Feature Entropy-Based | 15% | 55.2 | 52.7 | **44.9** | 50.9 | 94.1 |
> | Feature Magnitude-Based | 15% | 54.7 | 53.3 | 43.8 | 50.6 | 93.5 |
> | DyToK | 15% | **56.7** | **54.2** | 43.4 | **53.3** | **96.0** |
> | Attention Entropy-Based | 10% | 50.6 | 48.1 | 38.6 | 45.7 | 84.5 |
> | Feature Entropy-Based | 10% | 50.7 | 48.8 | 38.4 | 45.9 | 84.9 |
> | Feature Magnitude-Based | 10% | 50.6 | 47.4 | 37.9 | 45.3 | 83.7 |
> | DyToK | 10% | **53.2** | **50.4** | **42.8** | **48.8** | **90.2** |
>
> It can be observed that DyToK consistently outperforms entropy and magnitude-based methods across different retention ratios. We appreciate your valuable feedback and will include further analysis and discussion in the revised paper.
>
> &nbsp;
>
> > **Response to Question 2 (Definition of Keyframe and Adaptive Rate)**
> >
> - **Definition of Keyframe:** A keyframe is defined as a crucial frame that requires particular attention for accurate video understanding. For a comprehensive explanation, please refer to the Preliminary section (lines 94–101) and the cited literature there.
> - **Mechanism of Adaptive Token Allocation:** DyToK allocates initial token quantities per frame by multiplying frame weights from the temporal importance estimation stage with the specified token budget. It subsequently trims tokens based on an upper limit to mitigate temporal attention outliers (see Appendix B.1) and redistributes remaining tokens according to frame weights in descending order. Implementation details are described in Algorithm 1 of the Methodology section.
>
> We have completed additional visualizations that provide a clearer explanation of our method. As image attachments are not permitted in the rebuttal phase, these visualizations will be included in the revised paper.

---

> > ### Comment · Reviewer_HiJ7 · 2025-08-08
> >
> > Thank you to the authors for the detailed rebuttal. While the explanation of the attention-based importance estimation is helpful, the overall novelty of the proposed method remains limited. The lack of comparison with more recent state-of-the-art approaches—relying only on VisionZip and DyCoke as baselines—further restricts the demonstrated impact and potential for broader applicability. Given that some concerns remain unaddressed, I will maintain my original rating.

---

> ### Author Response · Authors · 2025-08-06
> **Look Forward to Feedback from Reviewer HiJ7**
>
> Dear Reviewer HiJ7,
>
> We sincerely appreciate the time and effort you devoted to reviewing our manuscript. In response to your thoughtful feedback, we have submitted a rebuttal with extensive experimental results and clarifications addressing your concerns, which includes the following key points:
>
> - **Comparison with SOTA Methods.** DyToK consistently outperforms recent CVPR 2025 SOTA methods, VisionZip and DyCoke, across six retention ratios and three long-video understanding benchmarks.
> - **Evaluation Against Structured Compression Baselines.** DyToK is a training-free token compression method for inference acceleration, thus complementary rather than competitive with techniques like LoRA or quantization that reduce training cost or require retraining.
> - **End-to-End Latency on Real Hardware.** DyToK is evaluated on VideoMME across retention ratios in terms of latency, FLOPs, and memory, and further validated on VideoChatGPT for long-sequence captioning, demonstrating strong accuracy-efficiency trade-offs.
> - **Justification for Attention-Based Importance Estimation.** Extensive ablations comparing DyToK’s cross-modal attention-based weighting with attention entropy-, feature entropy-, and feature magnitude-based methods confirm its superiority.
> - **Definition of Keyframe and Adaptive Rate.** We further clarify key concepts and provide additional visualizations for a clearer explanation of our method.
>
> We hope that these clarifications and additional experiments can address your concerns. Thank you again for your constructive review, and we welcome any further feedback or questions.
>
> Sincerely,
>
> Authors of Paper #19575

---

> ### Author Response · Authors · 2025-08-08
>
> Thank you for your responses and your acknowledgement. We would like to further clarify the following two aspects of our work:
>
> &nbsp;
>
> > **Comparison with the state-of-the-art methods**
>
> We have conducted extensive experiments using VisionZip and DyCoke, two representative and recently accepted methods at CVPR 2025 whose code and models are publicly available, enabling fair and reproducible comparisons. These baselines were selected because they were the most relevant accessible works accepted by CVPR 2025, **the latest top-tier conference** that had released its accepted list before the NeurIPS 2025 submission deadline (May 11, 2025).
>
> &nbsp;
>
> > **Impact and potential for broader applicability**
>
> We have conducted experiments with VisionZip and DyCoke on mainstream VLLMs (LLaVA-OneVision and Qwen2.5-VL) across six compression ratios (10%–75%), three long-video benchmarks, varying input frame lengths, and both encoder-feature and LLM-attention token compression paradigms. The results consistently demonstrate that our approach improves VLLMs’ inference efficiency in a **training-free** and **model-agnostic** manner, with **broad applicability** (see Appendix Tables 1–4, pp. 18–21 for full results).
>
> Moreover, as shown in discussions with other reviewers, our method exhibits:
>
> - **Cross-architecture transferability** (W2 of Reviewer mobj, W1/Q1 of Reviewer bZfF);
> - **Scalability across model sizes** (W3(a) of Reviewer nbsC);
> - **Strong generalization to different usage scenarios**, including fixed compute budgets, fixed input frames, and general descriptive queries (W1, Q2, Q4 of Reviewer bZfF).
> Importantly, the generalization capability of our designs can be supported by cross-model statistical analysis instead of relying on the empirical tuning (see our response to Reviewer bZfF’s follow-up), ensuring the method’s robustness in different settings.
>
> Given the above facts and analysis, we believe that the proposed DyTok is a simple, effective and general method. However, **its simplicity does not mean limited novelty**. To the best of our knowledge, no prior work has explored dynamic temporal budget allocation for accelerating the inference of VLLMs. DyTok introduces a **plug-and-play** mechanism that consistently improves inference efficiency across different existing models, **without any post-training overhead**.
>
> We hope this clarification highlights the method’s impact, broad applicability and novelty, and we sincerely appreciate your consideration.

---

### Note · Authors · 2025-08-14

We sincerely thank all reviewers for their constructive feedback and recognition of our work’s key contributions. For clarity, we refer to Reviewers HiJ7, nbsC, mobj, and bZfF as R1, R2, R3, and R4 in the following response.

In particular, we appreciate the reviewers’ recognition of DyToK’s *promising efficiency–accuracy tradeoffs* on long-video benchmarks (**R1-R4**). We are grateful that our finding that attention layers in VLLMs encode query-conditioned keyframe priors is viewed as *novel*, *insightful*, and *valuable for future research* (**R2, R4**). The acknowledgement of DyToK’s training-free, modular, plug-and-play design, enabling seamless integration with diverse backbones and token compression strategies, highlights its *broad applicability* (**R1-R4**). We are also encouraged that the paper is recognized as *clearly written*, *technically sound*, and *well-detailed* (**R2-R4**).

We have carefully addressed each comment and believe most concerns are resolved. Below, we summarize the additional experiments during rebuttal, which will be incorporated into the final version.

---
**Broader Applicability**
1. **Cross-Architecture Transferability:** Quantitative and statistical analyses show that DyToK remains robust with cross-architecture assistant–primary pairs.
2. **Broader Model Size:** Experiments on 3B, 7B, and 32B models reveal that VLLMs inherently possess keyframe priors, regardless of model size.
3. **General Descriptive Queries:** DyToK effectively identifies both semantic and content keyframes, with CLIP-based ablations further supporting the results.
4. **Fixed Input Frames / Computational Budget:** Under fixed 32-frame input and constrained token/FLOPs budgets, DyToK delivers notable efficiency and accuracy gains.

---
**Efficiency Analysis**
1. **End-to-End Latency on Real Hardware:** DyToK is evaluated on both multiple-choice and long-sequence captioning tasks across retention ratios in terms of latency, FLOPs, and memory.

---
**In-Depth Ablation**
1. **Importance Estimation Strategies:** DyToK’s cross-modal attention weighting outperforms entropy- and magnitude-based methods.
2. **Keyframe Extraction Strategies:** Compared with CLIP-based keyframe extraction in Video-RAG.
3. **Deep Layer Selection:** Experiments on various models confirm layer selection’s model-agnostic generalizability.
4. **Textual Token Selection:** Last-token and all-token strategies differ with retention, reflecting a precision–recall trade-off.

---

### Decision · Program_Chairs · 2025-09-17

**Decision:**

Accept (poster)

**Comment:**

This paper proposes a training-free, LLM-guided dynamic token compression for long-video VLLMs by exploiting attention-based keyframe priors; it offers plug-and-play compatibility with existing pruners and yields strong efficiency-accuracy trade-offs. Strengths are the novel insight, thorough ablations, cross-architecture transferability, and real-hardware latency results. Reviewer HiJ7 is the only reviewer who gives a negative opinion, mainly focusing on the lack of comparison experiments. During the rebuttal, authors have given sufficient comparison with VisionZip and dycoke, and end-to-end latency on real hardware. During the following discussion, reviewer HiJ7 did not give any response, which reduces the confidence of his/her rating.  Based on the above reasons, this paper is recommended for acceptance.